# Bridging the Gap Between Practice and PAC-Bayes Theory in Few-Shot Meta-Learning

**Nan Ding**
Google Research
dingnan@google.com

**Xi Chen**
Google Research
chillxichen@google.com

**Tomer Levinboim**
Google Research
tomerl@google.com

**Sebastian Goodman**
Google Research
seabass@google.com

**Radu Soricut**
Google Research
rsoricut@google.com

## Abstract

Despite recent advances in its theoretical understanding, there still remains a significant gap in the ability of existing PAC-Bayesian theories on meta-learning to explain performance improvements in the few-shot learning setting, where the number of training examples in the target tasks is severely limited. This gap originates from an assumption in the existing theories which supposes that the number of training examples in the observed tasks and the number of training examples in the target tasks follow the same distribution, an assumption that rarely holds in practice. By relaxing this assumption, we develop two PAC-Bayesian bounds tailored for the few-shot learning setting and show that two existing meta-learning algorithms (MAML and Reptile) can be derived from our bounds, thereby bridging the gap between practice and PAC-Bayesian theories. Furthermore, we derive a new computationally-efficient PACMAML algorithm, and show it outperforms existing meta-learning algorithms on several few-shot benchmark datasets.

## 1 Introduction

Recent advances in machine learning and neural networks have resulted in effective but parameter-bloated, data-hungry models. When the training data for a target task of interest is insufficient, such overparameterized models may easily overfit to the training data and exhibit poor generalization abilities. To address this problem, several research efforts have focused on designing a learning strategy that can leverage the training data of other tasks for the sake of improving the performance of some specific target task(s). Specifically, in the meta-learning (also called learning-to-learn or lifelong-learning) setting [6, 21], a meta-learner first extracts knowledge from a set of observed (meta-training) tasks and subsequently, this knowledge enables a base-learner to better adapt to the new, possibly data-limited target (meta-testing) task. The meta-learning framework has been successfully applied and made significant practical impact on computer vision [23], language understanding [7], reinforcement learning [9] and many other research fields.

In parallel to its impressive empirical success, a series of theoretical works [24, 18, 3, 22] study how meta-learning utilizes the knowledge obtained from the observed task data and how it generalizes to the unseen target task. Among the generalization bounds, PAC-Bayes bounds [15, 12] are considered especially tight and have already been proposed for meta-learning [18, 3, 22]. However, there still remains a gap between these existing PAC-Bayesian bounds and their practical application (especially in the few-shot setting), which originates from the assumption that the observed task environment $\tilde{T}$ and the target task environment $T$ are the same. In the PAC-Bayesian meta-learning setting, a task environment $T$ is a distribution from which $(D, m)$ is drawn from, where $D$ is the data distribution

35th Conference on Neural Information Processing Systems (NeurIPS 2021).

and $m$ is the number of training examples for the task. Although there is research work studying the case of general environment change (e.g. [19]) or data domain change (e.g. [11]), to the best of our knowledge, there is little work focusing on the case where only the number of training examples $\tilde{m}$ in the observed tasks and $m$ in the target task do not follow the same distribution. In practice, such mismatch commonly happens, because there is usually significantly more data in observed tasks than the target tasks, especially in the few-shot case. Without explicitly addressing this mismatch, the scope of the current theory is severely limited, and it prohibits a useful analysis on practical meta-learning algorithms such as MAML [9]. For example, when the number of training examples $m$ in the target task is small, the existing bounds yield a large generalization gap which grows with $O(1/m)$. In this paper, we bring the theory closer to practice by studying the setting where there are significantly more training examples in the observed task than in the target task (i.e., $\tilde{m} \gg m$). In Section 3.1, we study two practical meta-training strategies and provide their PAC-Bayesian bounds in Theorem 3 and Theorem 4. Both results are able to bring down the scaling coefficient of the bound from $O(1/m)$ to $O(1/\tilde{m})$. However, Theorem 3 introduces a penalty term in the bound that captures the discrepancy between the observed and target task environment. Motivated by MAML [9], we show with Theorem 4 that we can eliminate the penalty term by utilizing a subsampling strategy, yielding a much tighter bound.

This theoretical work also bridges the gap from practice to theory, as we further show that the maximum-a-posteriori (MAP) estimates of our bounds (in which the base-learner and the hyper-posterior are both approximated by Dirac-measures) yield various popular meta-learning algorithms, including multi-task pretraining [23], Reptile [16] and MAML [9]. In that sense, our PAC-Bayesian theories provide a different perspective for understanding and justifying these commonly used algorithms (Section 3.2).

Lastly, in Section 4, we propose PACMAML, a novel PAC-Bayesian meta-learning algorithm based on Theorem 4. As opposed to MAML, our algorithm does not have higher-order derivatives in the gradient, and therefore represents a significant improvement in computational efficiency. In Section 5, we conduct numerical experiments that empirically support the correctness of our theorems, and report the effectiveness of the new PACMAML algorithm, which obtains superior results on several few-shot benchmark datasets.

## 2 Preliminaries

We begin by reviewing the background and settings of the existing PAC-Bayesian bounds for meta-learning. Our notation mainly follows that of [22], which is itself adapted from [18, 3, 6].

**PAC-Bayesian for Supervised Learning** In supervised learning, a learning task is characterized by a data distribution $D$ over a data domain $Z$ where every example $z = (x, y)$. A hypothesis $h$ from the hypothesis space $H$ allows us to make predictions based on inputs $x$. The quality of the predictions is measured by a loss function $l(h, z)$, where the goal is to minimize the expected loss $L(h, D) = \mathbb{E}_{z \sim D} \, l(h, z)$. Typically, $D$ is unknown and instead we are given a set of $m$ observations $S \sim D^m = \{z_i \sim D\}_{i=1}^m$, in which case the empirical error on $S$ is simply $\hat{L}(h, S) = \frac{1}{m} \sum_{i=1}^m l(h, z_i)$.

In the PAC-Bayesian setting, we assume that the learner has prior knowledge of the hypothesis space $H$ in the form of a prior distribution $P(h)$. When the learner observes a training dataset $S$, it updates the prior into a posterior distribution $Q$. We formalize such a *base learner* $Q(S, P)$ that takes a dataset and a prior as input and outputs a posterior.

The expected error of the posterior $Q$ is called the Gibbs error $L(Q, D) = \mathbb{E}_{h \sim Q} \, L(h, D)$, and its empirical counterpart is $\hat{L}(Q, S) = \mathbb{E}_{h \sim Q} \, \hat{L}(h, S)$. The PAC-Bayesian framework provides the following bound over $L(Q, D)$ based on its empirical estimate $\hat{L}(Q, S)$.

**Theorem 1 ([2, 12])** *Given a data distribution $D$, a hypothesis space $H$, a prior $P$, a confidence level $\delta \in (0, 1]$, and $\beta > 0$, with probability at least $1 - \delta$ over samples $S \sim D^m$, we have for all posterior $Q$,*

$$L(Q, D) \leq \hat{L}(Q, S) + \frac{1}{\beta} \left( D_{KL}(Q \| P) + \log \frac{1}{\delta} \right) + \frac{m}{\beta} \Psi(\frac{\beta}{m}) \tag{1}$$

*where $\Psi(\beta) = \log \mathbb{E}_{h \sim P} \, \mathbb{E}_{z \sim D} \exp(\beta(l(h, z) - L(h, D)))$.*

**PAC-Bayesian for Meta-Learning**   In the meta-learning setting, the meta-learner observes different tasks $\tau_i = (D_i, m_i)$ during the meta-training stage, where all tasks share the same data domain $Z$, hypothesis space $H$ and loss function $l(h, z)$. For each observed task $\tau_i$, the meta-learner observes a training set $S_i$ of size $m_i$ which is assumed to be sampled i.i.d. from its respective data distribution $D_i$ (that is, $S_i \in D_i^{m_i}$). We further assume that each task $\tau_i = (D_i, m_i)$ is drawn i.i.d. from an environment $T$, which itself is a probability distribution over the data distributions and the sample sizes. The goal of meta-learning is to extract knowledge from the observed tasks $\tau_i$, which can then be used as prior knowledge for learning on new (yet unobserved) target tasks $\tau = (D, m) \sim T$. This prior knowledge is represented as a prior distribution $P(h)$ over learning hypotheses $h$, and it is subsequently used by the base learner $Q(S, P)$ for inference over the target tasks.

In the meta-learning PAC-Bayes framework, the meta-learner presumes a hyper-prior $\mathcal{P}(P)$ as a distribution over priors $P$. Upon observing datasets $S_1, \ldots, S_n$ from multiple tasks, the meta-learner updates the hyper-prior to a hyper-posterior $\mathcal{Q}(P)$. The performance of this hyper-posterior, also called the transfer-error, is measured as the expected Gibbs error when sampling priors $P$ from $\mathcal{Q}$ and applying the base learner:

$$R(\mathcal{Q}, T) := \mathbb{E}_{P \sim \mathcal{Q}} \, \mathbb{E}_{(D,m) \sim T} \, \mathbb{E}_{S \sim D^m} \left[ L(Q(S, P), D) \right]. \tag{2}$$

While $R(\mathcal{Q}, T)$ is unknown in practice, it can be estimated using the empirical error,

$$\hat{R}(\mathcal{Q}, S_{i=1}^n) := \mathbb{E}_{P \sim \mathcal{Q}} \left[ \frac{1}{n} \sum_{i=1}^n \hat{L}(Q(S_i, P), S_i) \right]. \tag{3}$$

In [18, 22], the following PAC-Bayesian meta-learning bound is provided:

**Theorem 2 ([18, 22])** *Given a task environment $T$ and a set of $n$ observed tasks $(D_i, m_i) \sim T$, let $\mathcal{P}$ be a fixed hyper-prior and $\lambda > 0$, $\beta > 0$, with probability at least $1 - \delta$ over samples $S_1 \in D_1^{m_1}, \ldots, S_n \in D_n^{m_n}$, we have, for all base learner $Q$ and all hyper-posterior $\mathcal{Q}$,*

$$R(\mathcal{Q}, T) \leq \hat{R}(\mathcal{Q}, S_{i=1}^n) + \left( \frac{1}{\lambda} + \frac{1}{n\beta} \right) D_{KL}(\mathcal{Q} \| \mathcal{P})$$

$$+ \frac{1}{n\beta} \sum_{i=1}^n \mathbb{E}_{P \sim \mathcal{Q}} \left[ D_{KL}(Q(S_i, P) \| P) \right] + C(\delta, \lambda, \beta, n, m_i). \tag{4}$$

Here $C(\delta, \lambda, \beta, n, m_i)$ contains $\Psi$ and $\frac{1}{\delta}$ terms as in Eq.(1) (see Appendix A.1), and can be bounded by a function that is independent of $\mathcal{Q}$ for both bounded and unbounded loss functions under moment constraints (see details in [22]). From a Bayesian perspective, meta-learning attempts to learn a good hyper-posterior $\mathcal{Q}$ such that for all tasks in the task environment $T$, the divergence terms $D_{KL}(Q(S_i, P) \| P)$ would be substantially smaller in expectation when $P \sim \mathcal{Q}$ compared to when $P \sim \mathcal{P}$, such as in the ordinary supervised learning setting of Eq.(1).

The hyperparameters $\lambda$ and $\beta$ can be adjusted to balance between the first three terms of the bound and the $C$ function. Defining the harmonic mean of $m_i$ as $\tilde{m} = (\sum_{i=1}^n 1/nm_i)^{-1}$, a common choice is $\lambda \propto n$ and $\beta \propto \tilde{m}^*$. In this case, the generalization gap $R(\mathcal{Q}, T) - \hat{R}(\mathcal{Q}, S_{i=1}^n)$ becomes at least $O(\frac{1}{\tilde{m}})$ (from the 3rd-term on the RHS of Eq.4). In the next section, we examine an assumption in this bound which makes it impractical for the few-shot setting.

## 3   Bridging the Gap between Practice & Theory of Few-Shot Meta-Learning

The previous PAC-Bayesian meta-learning bound (Theorem 2) assumes that the number of training examples $m_i$ for the observed tasks $\tau_i$ and the number of training examples $m$ for the target task $\tau$ are drawn from the same distribution (i.e. $\mathbb{E}_T[m_i] = \mathbb{E}_T[m]$). However, practical applications of meta-learning such as [23, 7] operate in a setting where there are far more training examples in the observed tasks than in the target task. Moreover, focusing on the few-shot setting (where $m$ is particularly small) exposes a gap between theory and practice – Theorem 2 is unable to use the large number of observed samples and can only produce a loose bound of $O(\frac{1}{m})$ which is ineffective at explaining the impressive generalization performance of meta-learning as reported in practice.

---

*Another common choice is $\lambda \propto \sqrt{n}$ and $\beta \propto \sqrt{\tilde{m}}$, so that the bound is asymptotically consistent, and scales with $O(\frac{1}{\sqrt{\tilde{m}}})$. However, in practice the bound with $\beta \propto \tilde{m}$ is usually tighter [10].

In this section we attempt to close this gap by deriving an effective PAC-Bayesian bound (Theorem 4) tailored for the few-shot setting. Interestingly, the bounds derived in this section also provide PAC-Bayesian justifications for two practical algorithms, Reptile and MAML.

## 3.1 Practical PAC-Bayesian Bounds for Few-Shot Meta-Learning

A first attempt at leveraging the larger number of examples $m_i$ in the observed tasks is to directly follow the learning strategy of Theorem 2, by bounding $R(\mathcal{Q}, T)$ using the empirical risk $\hat{R}(\mathcal{Q}, S_{i=1}^n)$, with $S_i \in D_i^{m_i}$ and $(D_i, m_i) \sim \tilde{T}$, despite the change of task environment from $T$ to $\tilde{T}$. This slight generalization leads to the following bound (with proof in Appendix A.2):

**Theorem 3** *For a target task environment $T$ and an observed task environment $\tilde{T}$ where $\mathbb{E}_{\tilde{T}}[D] = \mathbb{E}_T[D]$ and $\mathbb{E}_{\tilde{T}}[m] \geq \mathbb{E}_T[m]$, let $\mathcal{P}$ be a fixed hyper-prior and $\lambda > 0$, $\beta > 0$, then with probability at least $1 - \delta$ over samples $S_1 \in D_1^{m_1}, \dots, S_n \in D_n^{m_n}$ where $(D_i, m_i) \sim \tilde{T}$, we have, for all base learners $Q$ and hyper-posterior $\mathcal{Q}$,*

$$R(\mathcal{Q}, T) \leq \hat{R}(\mathcal{Q}, S_{i=1}^n) + \left( \frac{1}{\lambda} + \frac{1}{n\beta} \right) D_{KL}(\mathcal{Q} \,\|\, \mathcal{P})$$

$$+ \frac{1}{n\beta} \sum_{i=1}^n \mathbb{E}_{P \sim \mathcal{Q}} \left[ D_{KL}(Q(S_i, P) \| P) \right] + C(\delta, \lambda, \beta, n, m_i) + \Delta_\lambda(\mathcal{P}, T, \tilde{T}), \qquad (5)$$

*where $\Delta_\lambda(\mathcal{P}, T, \tilde{T}) = \frac{1}{\lambda} \log \mathbb{E}_{P \in \mathcal{P}} \, e^{\lambda(R(P, T) - R(P, \tilde{T}))}$.*

When $\mathbb{E}_{\tilde{T}}[m_i] \gg \mathbb{E}_T[m]$, this decoupling of the task environments seems beneficial at first, because $O(\frac{1}{\tilde{m}})$ is smaller compared to Eq.(4) when $\beta \propto \tilde{m}$. Unfortunately however, Eq.(5) introduces an additional penalty term $\Delta_\lambda$, which increases as $\mathbb{E}_{\tilde{T}}[\tilde{m}]$ gets larger.

To understand the influence of $\Delta_\lambda$, we plot the (blue) bound of Eq.(5) in Fig.1 by using the synthetic Sinusoid regression task (see details in Section 5.1 and in Appendix D.4) where we fixed $m = 5$ and varied $m_i$ from 5 to 100. When $m_i = m = 5$, Eq.(5) reduces to Eq.(4) and $\Delta_\lambda = 0$. Contrary to intuition, increasing $m_i$ does not reduce the bound, but instead makes it worse due to the rapid increase of $\Delta_\lambda$.

Can we utilize more training examples without introducing a penalty term such as $\Delta_\lambda$? In the definition of $\hat{R}(\mathcal{Q}, S_{i=1}^n)$ (Eq.(3)), we note that the training dataset $S_i$ of the observed task $\tau_i$ is used twice: first in training the base-learner $Q(S_i, P)$, and then, in evaluating the empirical risk $\hat{L}(Q, S_i)$. In analyzing the proof of the theorem (see Appendix A.2), it can be seen that the penalty term arises exactly because $Q(S_i, P)$ is trained over more samples compared to $Q(S, P)$ of the target task, which results in the more powerful base-learners during meta-training than the one for the target task.

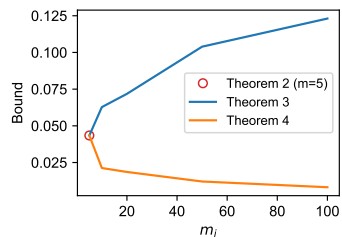

Figure 1: The PAC-Bayesian bounds of Theorems 2, 3, & 4 as evaluated over the Sinusoid dataset. Some constant terms are neglected (see Appendix D.4 for more details).

This motivates us to develop a MAML-inspired learning strategy, in which we maintain the same target-task training environment $T$ for the base-learners of the observed tasks: we first sample a subset $S_i' \in D_i^{m_i'}$ from $S_i$ where $m_i'$ and $m$ follow the same distribution and $m_i' \leq m_i$. Then we use only the subset $S_i'$ to train the base-learner $Q(S_i', P)$. At the same time, all examples of $S_i \in D_i^{m_i}$ are used for evaluating the empirical risk $\hat{L}(Q, S_i)$, so that the larger $m_i$ in the empirical risk $\hat{L}(Q, S_i)$ help tightening the generalization gap. This new strategy leads to the following bound (proof in Appendix A.3):

**Theorem 4** *For a target task environment $T$ and an observed task environment $\tilde{T}$ where $\mathbb{E}_{\tilde{T}}[D] = \mathbb{E}_T[D]$ and $\mathbb{E}_{\tilde{T}}[m] \geq \mathbb{E}_T[m]$, let $\mathcal{P}$ be a fixed hyper-prior and $\lambda > 0$, $\beta > 0$, then with probability at least $1 - \delta$ over samples $S_1 \in D_1^{m_1}, \dots, S_n \in D_n^{m_n}$ where $(D_i, m_i) \sim \tilde{T}$, and subsamples $S_1' \in D_1^{m_1'} \subset S_1, \dots, S_n' \in D_n^{m_n'} \subset S_n$, where $\mathbb{E}[m_i'] = \mathbb{E}_T[m]$, we have, for all base learner $Q$*

*and all hyper-posterior $\mathcal{Q}$,*

$$R(\mathcal{Q}, T) \leq \mathbb{E}_{P \sim \mathcal{Q}} \left[ \frac{1}{n} \sum_{i=1}^{n} \hat{L}(Q(S'_i, P), S_i) \right] + \left( \frac{1}{\lambda} + \frac{1}{n\beta} \right) D_{KL}(\mathcal{Q} \,\|\, \mathcal{P})$$

$$+ \frac{1}{n\beta} \sum_{i=1}^{n} \mathbb{E}_{P \sim \mathcal{Q}} \left[ D_{KL}(Q(S'_i, P) \| P) \right] + C(\delta, \lambda, \beta, n, m_i). \tag{6}$$

This bound is still $O(\frac{1}{\tilde{m}})$ when choosing $\beta \propto \tilde{m}$, but unlike Eq.(5), it does not have an additional penalty term in Eq.(6), which is due to the shared training environment $T$ of the base-learners in both observed and target tasks. Importantly, the resulting bound is effective in the few-shot setting as an increase in the number of observed examples $m_i$ monotonically tightens the generalization gap. This is visually demonstrated in Fig.1 in which the (orange) bound of Eq.(6) in Theorem 4 is monotonically decreasing as $m_i$ increases, while the bound in Theorem 2 is limited only to $m_i = 5$ and the bound of Theorem 3 grows.

### 3.2 Justifying Reptile and MAML using PAC-Bayesian Bounds

It is worth noting that Theorems 3 and 4 not only address more practical scenarios in which observed (meta-training) examples are more abundant than the target examples, but they also serve as a justification for popular and practical meta-learning algorithms: Reptile [16] and MAML [9].

To show this, let us consider the maximum-a-posteriori (MAP) approximations on the hyper-posterior $\mathcal{Q}(P)$ and base-leaner $Q_i(h), \forall i = 1, \ldots, n$, with Dirac measures. In addition, we use the isotropic Gaussian priors with variance hyperparameter $\sigma_0^2$ and $\sigma^2$ for the hyper-prior $\mathcal{P}(P)$ and the prior $P(h)$. The hypothesis $h$ is parameterized by $\mathbf{v}$. Then we have

$$\mathcal{P}(P) = \mathcal{N}(\mathbf{p} \,|\, 0, \sigma_0^2), \ \mathcal{Q}(P) = \delta(\mathbf{p} = \mathbf{p}_0), \ P(h_\mathbf{v}) = \mathcal{N}(\mathbf{v} \,|\, \mathbf{p}, \sigma^2), \ Q_i(h_\mathbf{v}) = \delta(\mathbf{v} = \mathbf{q}_i),$$

and the goal of MAP approximation is to find the optimal meta-parameters $\mathbf{p}_0$. With the above assumptions, the PAC-Bayesian bound (denoted PacB) of Eq.(5) and Eq.(6) with respect to $\mathbf{p}_0$ becomes (up to a constant, see Appendix B),

$$PacB(\mathbf{p}_0) = \frac{1}{n} \sum_{i=1}^{n} \hat{L}(\mathbf{q}_i, S_i) + \frac{\tilde{\xi} \|\mathbf{p}_0\|^2}{2\sigma_0^2} + \frac{1}{n\beta} \sum_{i=1}^{n} \frac{\|\mathbf{p}_0 - \mathbf{q}_i\|^2}{2\sigma^2}, \tag{7}$$

where $\tilde{\xi} = \frac{1}{\lambda} + \frac{1}{n\beta}$. Here, $\mathbf{q}_i$ can be any function of $\mathbf{p}_0$ and $S_i$ for Eq.(5) (or $\mathbf{p}_0$ and $S'_i$ for Eq.(6)), such that the only free variable in Eq.(7) is $\mathbf{p}_0$. Indeed, by setting $\mathbf{q}_i$ according to the choices below, we can derive the gradients of several meta-learning algorithms.

When $\mathbf{q}_i = \mathbf{p}_0$, the gradient of Eq.(7) reduces to that of multi-task pretraining [23, 7],

$$\lim_{\mathbf{q}_i \to \mathbf{p}_0} \frac{d(PacB)}{d\mathbf{p}_0} = \frac{\tilde{\xi} \mathbf{p}_0}{\sigma_0^2} + \frac{1}{n} \sum_{i=1}^{n} \frac{d}{d\mathbf{p}_0} \hat{L}(\mathbf{p}_0, S_i).$$

On the other hand, if we use the optimal Dirac-base-learner $\mathbf{q}_i^*$ of $\mathbf{p}_0$ and $S_i$, such that

$$\mathbf{q}_i^* = \operatorname*{argmin}_{\mathbf{q}_i} \left[ \hat{L}(\mathbf{q}_i, S_i) + \frac{\|\mathbf{p}_0 - \mathbf{q}_i\|^2}{2\beta\sigma^2} \right], \tag{8}$$

then the gradient of Eq.(7) becomes substantially simpler (see details in the Appendix B),

$$\frac{d(PacB)}{d\mathbf{p}_0} = \frac{\tilde{\xi} \mathbf{p}_0}{\sigma_0^2} + \frac{1}{n} \sum_{i=1}^{n} \frac{\mathbf{p}_0 - \mathbf{q}_i^*}{\beta\sigma^2}, \tag{9}$$

and in fact, Eq.(9) is equivalent to the meta-update rule of the Reptile algorithm [16], whose inner-loop is an approximate algorithm for solving the optimal Dirac-base-learner $\mathbf{q}_i^*$.

Lastly, when $\mathbf{q}_i$ is a few gradient descent steps of $\hat{L}(\mathbf{q}_i, S'_i)$ with initial $\mathbf{q}_i = \mathbf{p}_0$, the gradient of Eq.(7) reduces to that of the MAML algorithm[†] [9] as $\sigma^2 \to \infty$,

$$\lim_{\sigma^2 \to \infty} \frac{d(PacB)}{d\mathbf{p}_0} = \frac{\tilde{\xi} \mathbf{p}_0}{\sigma_0^2} + \frac{1}{n} \sum_{i=1}^{n} \frac{d}{d\mathbf{p}_0} \hat{L}(\mathbf{q}_i, S_i).$$

---

[†]A slight difference is that MAML usually assumes $S_i \cap S'_i = \emptyset$; while in our setting, we assume $S'_i \subset S_i$. However, Theorem 4 still holds when $S_i \cap S'_i = \emptyset$.

One observation here is that, since $\mathbf{q}_i$ is function of the gradient of $\mathbf{p}_0$, $d\,\mathbf{q}_i\,/d\,\mathbf{p}_0$ involves high-order gradient w.r.t. $\mathbf{p}_0$, which would result in a computationally intensive algorithm. In the next section we present a computationally efficient algorithm which relies only on first-order derivatives.

## 4  PAC-Bayesian Meta-Learning Algorithms in the Few-Shot Setting

In this section we present two PAC-Bayesian based Meta-Learning algorithms with non-Dirac base-learners. We first derive their objective functions from the RHS of Eq.(5) and Eq.(6), and then derive low-variance gradient estimators for their optimization.

First, since Eq.(4) and Eq.(5) only differ by $\Delta_\lambda$, we follow [22] and plug in their proposed Gibbs posterior based base-learner $Q^*(S_i, P)(h) = P(h)\exp(-\beta\hat{L}(h, S_i))/Z_\beta(S_i, P)$ into Eq.(5), which minimizes Eq.(5) w.r.t. $Q$. This yields that, with at least $1 - \delta$ probability,

$$R(\mathcal{Q}, T) \leq \frac{1}{n}\sum_{i=1}^{n}\mathbb{E}_{P\sim\mathcal{Q}}\underbrace{\left[-\frac{1}{\beta}\log Z_\beta(S_i, P)\right]}_{W_1} + \tilde{\xi}D_{KL}(\mathcal{Q}\,\|\,\mathcal{P}) + \Delta_\lambda + C \tag{10}$$

where $\tilde{\xi} = \frac{1}{\lambda} + \frac{1}{n\beta}$ and $C$ is the same constant from the previous bounds. Since $\Delta_\lambda$ is independent of $\mathcal{Q}$ and can be neglected during inference or optimization of $\mathcal{Q}$, it reduces to the same PACOH objective as in [22].

On the other hand, the same Gibbs posterior cannot be used as the base learner of Eq.(6), because the Gibbs posterior would depend on $S_i$, while the base learner in Eq.(6) should only be dependent on $S_i' \subset S_i$. Therefore, we use the following posterior $Q_i^\alpha$ with hyperparameter $\alpha$,

$$Q_i^\alpha(S_i', P)(h) = \frac{P(h)\exp(-\alpha\hat{L}(h, S_i'))}{Z_\alpha(S_i', P)}.$$

Plugging into Eq.(6) (derivations in Appendix) yields that, with at least $1 - \delta$ probability,

$$R(\mathcal{Q}, T) \leq \frac{1}{n}\sum_{i=1}^{n}\mathbb{E}_{P\sim\mathcal{Q}}\underbrace{\left[-\frac{1}{\beta}\log Z_\alpha(S_i', P) + \hat{L}\frac{\Delta}{\frac{\alpha}{\beta}}(Q_i^\alpha, S_i, S_i')\right]}_{W_2} + \tilde{\xi}D_{KL}(\mathcal{Q}\,\|\,\mathcal{P}) + C. \tag{11}$$

where $\hat{L}\frac{\Delta}{\frac{\alpha}{\beta}}(Q_i^\alpha, S_i, S_i') \triangleq \hat{L}(Q_i^\alpha, S_i) - \frac{\alpha}{\beta}\hat{L}(Q_i^\alpha, S_i')$. We refer to the RHS of this equation as the PACMAML objective, because Eq.(11) comes from the PAC-Bayesian bound of Eq.(6), which is similar to MAML in subsampling the training data for base-learners.

Given these two objectives, the next step is to estimate the gradients of $W_1$ and $W_2$, which can then be plugged into Monte-Carlo methods for estimating a hyper-posterior distribution of $\mathcal{Q}$ (or optimization methods for finding an MAP solution).

**Gradient Estimation**    In $W_1$ and $W_2$, the terms $Z_\beta, Z_\alpha, \hat{L}\frac{\Delta}{\frac{\alpha}{\beta}}(Q_i^\alpha, S_i, S_i')$ all involve integrations over $h$. When $P(h)$ is Gaussian and $\hat{L}(h, S_i)$ is a squared loss, such integrations have closed form solutions and the gradients can be analytically obtained. However, when $\hat{L}(h, S_i)$ is not a squared loss (such as the softmax loss), the integration does not have a closed form solution and we resort to approximations. For example, [22] directly approximates the objective $W_1$ with Monte-Carlo sampling, which however results in a biased gradient estimator.

Here, we follow an alternative approach from the REINFORCE algorithm [28], which instead approximates *the gradient of the objective* with Monte-Carlo methods, and has the benefit that the resulting gradient estimator is unbiased. Assuming that the model hypothesis $h$ is parameterized by $\mathbf{v}$ such that $\hat{L}(h, S_i) \triangleq \hat{L}(\mathbf{v}, S_i)$, and $\mathbf{v}$ has prior $P(\mathbf{v}) = \mathcal{N}(\mathbf{v}\,|\,\mathbf{p}, \sigma^2)$ with meta-parameter $\mathbf{p}$, then

$$\log Z_\beta(S_i, \mathbf{p}) = \log\int\mathcal{N}(\mathbf{v}\,|\,\mathbf{p}, \sigma^2)\exp(-\beta\hat{L}(\mathbf{v}, S_i))d\,\mathbf{v}\,.$$

Note that $\mathbf{p}$ appears in the probability distribution $\mathcal{N}(\mathbf{v}\,|\,\mathbf{p}, \sigma^2)$ of the expectation, and the naive Monte-Carlo estimator of the gradient w.r.t. $\mathbf{p}$ is known to exhibit high variance. To reduce the

variance, we apply the reparameterization trick [13] and rewrite $\mathbf{v} = \mathbf{p} + \mathbf{w}$ with $\mathbf{w} \sim \mathcal{N}(\mathbf{w} \,|\, \mathbf{0}, \sigma^2)$. This leads to the following gradient of $W_1$,

$$\frac{dW_1}{d\mathbf{p}} = -\frac{1}{\beta}\frac{d}{d\mathbf{p}}\log Z_\beta(S_i, \mathbf{p}) = \int Q_i^\beta(\mathbf{w}; S_i)\frac{\partial \hat{L}(\mathbf{p} + \mathbf{w}, S_i)}{\partial \mathbf{p}}d\mathbf{w}, \qquad (12)$$

$$\text{where, } Q_i^\beta(\mathbf{w}; S_i) \propto \mathcal{N}(\mathbf{w} \,|\, \mathbf{0}, \sigma^2)\exp(-\beta\hat{L}(\mathbf{p} + \mathbf{w}, S_i)).$$

As for $W_2$, we also need to evaluate the gradient of $\hat{L}_{\frac{\alpha}{\beta}}^\Delta(Q_i^\alpha, S_i, S_i')$, where

$$\frac{d}{d\mathbf{p}}\hat{L}_{\frac{\alpha}{\beta}}^\Delta(Q_i^\alpha, S_i, S_i') = \int Q_i^\alpha(\mathbf{w}; S_i')\frac{\partial \hat{L}_{\frac{\alpha}{\beta}}^\Delta(\mathbf{p} + \mathbf{w}, S_i, S_i')}{\partial \mathbf{p}}d\mathbf{w} + \int \frac{\partial Q_i^\alpha(\mathbf{w}; S_i')}{\partial \mathbf{p}}\hat{L}_{\frac{\alpha}{\beta}}^\Delta(\mathbf{p} + \mathbf{w}, S_i, S_i')d\mathbf{w}. \tag{13}$$

The first term of Eq.(13) is similar to the gradient in Eq.(12). The Monte-Carlo gradient estimator of the second term, however, exhibits the same high-variance problem as in the policy gradient method. As a remedy, we approximate the gradient with the one from the Softmax Policy Gradient [8], which yields a low-variance approximate gradient of $W_2$ (details in Appendix):

$$\frac{dW_2}{d\mathbf{p}} \simeq \int Q_i^\alpha(\mathbf{w}; S_i')\frac{\partial \hat{L}(\mathbf{p} + \mathbf{w}; S_i)}{\partial \mathbf{p}}d\mathbf{w} + \frac{\alpha}{\beta}\int \left(Q_i^\beta(\mathbf{w}; S_i) - Q_i^\alpha(\mathbf{w}; S_i')\right)\frac{\partial \hat{L}(\mathbf{p} + \mathbf{w}; S_i')}{\partial \mathbf{p}}d\mathbf{w}. \tag{14}$$

The first-term in Eq.(14) is similar to the gradient of the First-order MAML (FOMAML, [9]). The second term involves $Q_i^\beta$ and $Q_i^\alpha$, which are similar to the leader and the chaser in BMAML [29]. Intuitively, the second term provides additional information that plays a similar role to the high-order derivatives in MAML. However, unlike MAML and BMAML, Eq.(14) only involves partial derivatives over $\mathbf{p}$ (since $\mathbf{w}$ is not a function of $\mathbf{p}$) and therefore relies only on first-order derivatives which contribute to its efficiency and stability.

To estimate Eq.(12) and Eq.(14) in practice, we first draw samples $\mathbf{w}_{(n)}^\alpha \sim Q_i^\alpha(\mathbf{w}; S_i')$ and $\mathbf{w}_{(n)}^\beta \sim Q_i^\beta(\mathbf{w}; S_i)$ using the Monte-Carlo sampling (e.g. SGLD [27] or SVGD [14]). After plugging the samples into $\hat{L}(\mathbf{p} + \mathbf{w}; S_i)$ and $\hat{L}(\mathbf{p} + \mathbf{w}; S_i')$, we can apply automatic gradient computations (with Tensorflow [1] or Pytorch [17]) over $\mathbf{p}$ to get the stochastic gradient estimator of $W_1$ and $W_2$.

## 5  Experiments

In this section, we evaluate the two PAC-Bayesian algorithms as they were derived in the previous section: PACOH [22] of Eq.(10) and PACMAML of Eq.(11). We use several few-shot learning benchmarks (both synthetic and real), and compare them against other existing meta-learning algorithms, including MAML [9], Reptile [16], and BMAML [29]. To fairly compare with other meta-learning algorithms that optimize a single model, we consider only the empirical Bayes method for PACOH and PACMAML, in which a single MAP solution of $\mathcal{Q}$ is used, instead of Bayesian ensembles of $\mathcal{Q}$.

### 5.1  Few-Shot Regression Problem

Our first set of experiments are based on the synthetic regression environment setup from [22], where the gradient can be obtained analytically. The base-learners $Q(S, P)$ are modeled using Gaussian Process (GP) regression with a prior $P_\theta(h) = \mathcal{GP}(h|m_\theta(x), k_\theta(x, x'))$, where the mean function $m_\theta$ and the kernel function $k_\theta$ are instantiated as neural networks as in [22]. For every example $z_j = (x_j, y_j)$ and a hypothesis $h$, the loss function is $l(h, z_j) = \|h(x_j) - y_j\|_2^2$ and the empirical risk is $\hat{L}(h, S_i) = \frac{1}{m_i}\sum_{j=1}^{m_i} l(h, z_j)$. The hyper-prior $\mathcal{P}(P_\theta) := \mathcal{P}(\theta) = \mathcal{N}(\theta|0, \sigma_0^2 I)$ is an isotropic Gaussian defined over the network parameters $\theta$. The MAP approximated hyper-posterior takes the form of a delta function, where $\mathcal{Q}_{\theta_0}(P_\theta) := \mathcal{Q}_{\theta_0}(\theta) = \delta(\theta = \theta_0)$. As a result, we have that $D_{KL}(\mathcal{Q}_{\theta_0} \,\|\, \mathcal{P}) = \|\theta_0\|^2/2\sigma_0^2$, where we use $\sigma_0^2 = 3$ in our experiments.

We experiment with the synthetic *Sinusoid* environment (details in Appendix D.2), where we fix the number of observed tasks $n = 20$, and vary the number of examples per observed tasks from $m_i \in \{5, 10, 30, 50, 100\}$. The number of training examples for each target task is fixed to be $m = 5$, and another 100 examples for each target task are used as a test set to evaluate the generalization error. We report the averaged generalization error over 40 models, with the hyperparameters selected

by 4-fold cross-validation over the 20 target tasks. Each model is trained on 1 of the 8 pre-sampled meta-training sets (each containing $n = 20$ observed tasks) and each set is run with 5 random seeds for network initialization. $\alpha$ and $\beta$ are chosen based on the cross-validation from the grid $\beta/m_i \in \{10, 30, 100\}$, and $\alpha/m_i \in \{10, 20, 30, 40, 50, 60\}$.

Figure 2 shows the averaged generalization errors (RMSE) as $m_i$ changes, for the Reptile (with optimal $\mathbf{q}_i^*$), MAML, PA-COH, and PACMAML algorithms. The size of $S_i'$ used for base-learner training in MAML and PACMAML is $m_i' = 5$ for all $m_i$. The hyperparameter values, the validation errors and the standard errors are reported in the Appendix D.3. As can be seen from the figure, the generalization errors of Reptile (blue) and PACOH (green), both derived from Theorem 3, have a U-shaped curve. That is, increasing the meta-training data $m_i$ initially improves generalization in the few-shot target tasks, however, as $m_i$ continues to grows well beyond $m$, generalization suffers. This confirms our conjecture from Theorem 3, that larger $m_i$ has a mixed effect on its generalization behavior due to the penalty term $\Delta_\lambda$. In contrast, the generalization error of MAML and PACMAML, both derived from Theorem

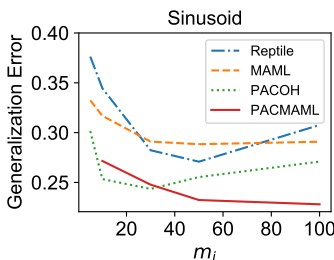

Figure 2: Generalization error (RMSE) on the Sinusoid dataset: PACMAML and MAML continue to improve as $m_i$ increases.

4, is monotonically decreasing as desired. Both the generalization error and the bound (in Fig.1) demonstrate that PACMAML is the most effective strategy of utilizing larger meta-training data for few-shot learning.

## 5.2 Few-shot Classification Problems

In addition to the regression problems where the gradients have closed-form, our next experiments evaluate how PACMAML perform on classification tasks using softmax losses with gradient estimators from Eq.(12) and Eq.(14). In order to fairly compare with MAML, which has only one set of inner adaptive parameters, we also only use one sample for approximating the inner posterior distribution $\mathcal{Q}_i^\alpha$ and $\mathcal{Q}_i^\beta$.

**Image Classification**  Our first classification experiment is based on the miniImagenet classification task [25] involving a task adaptation of 5-way classification with a single training example per class (1-shot). The dataset consists of 60,000 color images of 84×84 dimension. The examples consist of total 100 classes that are partitioned into 64, 12, and 24 classes for meta-train, meta-validation, and meta-test, respectively. We generated the tasks following the same procedure as in [9] and used the same feature extraction model which contains 4 convolutional layers. Although the original MAML adapted the entire network in the inner loop, [20] showed similar results by adapting only the top layer, which significantly reduced computational complexity. We follow the same "almost no inner loop" (ANIL) setting as [20], and compare MAML with BMAML, PACOH and PACMAML. Reptile is not included, because it requires full model adaptation.

For all algorithms, we optimize for 6 steps in the inner loop to obtain the inner adaptive parameter (or a posterior sample $\mathbf{w}$). The data sizes of the observed tasks are varied from $m_i = \{10, 20, 40, 80\}$ and $m_i' = m = 5$ (one shot for each of 5 classes). We fixed $\alpha/\beta = m_i'/m_i$ and perform grid search on $\alpha$ as well as the meta and inner learning rate on the meta-validation dataset. Other hyperparameters followed the setting in [9]. Further details are reported in the Appendix.

|  | FOMAML | MAML | BMAML | PACOH | PACMAML |
|---|---|---|---|---|---|
| $m_i = 10$ | $41.8 \pm 0.9$ | $47.3 \pm 0.9$ | $29.9 \pm 0.9$ | $31.2 \pm 0.8$ | $\mathbf{47.8 \pm 0.9}$ |
| $m_i = 20$ | $44.3 \pm 0.9$ | $48.0 \pm 0.9$ | $34.3 \pm 0.9$ | $37.0 \pm 0.9$ | $\mathbf{49.1 \pm 0.9}$ |
| $m_i = 40$ | $46.2 \pm 1.0$ | $47.8 \pm 0.9$ | $41.5 \pm 0.9$ | $41.6 \pm 0.9$ | $\mathbf{48.9 \pm 0.9}$ |
| $m_i = 80$ | $45.7 \pm 0.9$ | $48.1 \pm 0.9$ | $44.2 \pm 0.9$ | $44.6 \pm 0.9$ | $\mathbf{50.1 \pm 0.9}$ |

Table 1: Averaged test accuracy and standard error in the ANIL setting.

The main meta-testing results are presented in Table 1. We find that PACOH performs significantly worse than PACMAML. One explanation for this is that in PACOH the base-learner (for top layer) is exposed to all $S$ data, and may have already overfit on $S$ and the meta-learner (for lower layers) is unable to learn further. The overfitting of the base-learner is more severe when $m_i$ is small.

Surprisingly, we find that BMAML behaves similarly poor in the ANIL 1-particle setting. In FOMAML, MAML and PACMAML, the base-learner is only trained on $S'$ and the meta-learner can learn from the unseen examples in $S$ and therefore no overfitting happens. Both MAML and PACMAML performs significantly better than FOMAML when $m_i$ is small, but their performances saturate and improve little for larger $m_i$, which may due to the domain change between meta-training and testing (as the image class changes). Overall, PACMAML as a first-order method not only significantly outperforms FOMAML, but also marginally outperforms the high-order MAML, which demonstrates the effectiveness of PACMAML and its gradient estimator.

**Natural Language Inference** Lastly, we evaluate the meta-learning algorithms on the large-scale BERT-base [7] model containing 110M parameters. Our experiment involves 12 practical natural language inference tasks from [4] which include:[‡] (1) entity typing: CoNLL-2003, MIT-Restaurant; (2) rating classification: the review ratings from the Amazon Reviews dataset in the domain of Books, DVD, Electronics, Kitchen; (3) text classification: social-media datasets from crowdflower that include Airline, Disaster, Emotion, Political Bias, Political Audience, Political Message.

Following [4], we used the pretrained BERT-base model as our base model (hyper-prior), and used GLUE benchmark tasks [26] for meta-training the models and meta-validation for hyperparameter search, before fine-tuning them for the 12 target tasks. The fine-tuning data contains $k \in \{4, 8, 16\}$-shot data for each class in each task. For every $k$, 10 fine-tuning datasets were sampled for each target task. The final reported result is the average of the 10 models fine-tuned over these 10 datasets (for each task and each $k$ separately), and evaluated on the entire test set for each target task [4]. The data size of the observed tasks are fixed to be $m_i = 256$, where the data points for each observed task are randomly sampled from the training data of one of the GLUE tasks. Because the number of classes in these 12 few-shot tasks varies from 2 to 12, we choose the inner data size $m_i'$ from $\{32, 64\}$ for MAML, BMAML and PACMAML. As in [4], we also partition the set of model parameters to task-specific and task-agnostic. For the 12-layer BERT-base model, we consider a hyper-parameter $v \in \{6, 9, 11, 12\}$, where only the layers higher than the $v$-th layer are considered task-specific and will be adapted in the inner loop. When $v = 12$, only the top classification layers are adaptable. For BMAML, PACOH and PACMAML, we performed grid search on $\alpha$ and fixed $\alpha/\beta = m_i'/m_i$.

| $k$ | H-SMLMT [5] | MAML | BMAML | PACOH | PACMAML |
|---|---|---|---|---|---|
| 4 | 48.61 | 48.21 | 47.27 | 50.47 | **51.58** |
| 8 | 52.92 | 53.52 | 52.08 | 54.83 | **55.68** |
| 16 | 57.90 | 57.38 | 56.53 | 58.22 | **59.18** |

| | $v$=6 | $v$=9 | $v$=11 | $v$=12 |
|---|---|---|---|---|
| MAML | 120G | 57G | 16G | 4G |
| BMAML | 121G | 59G | 19G | 4G |
| PACMAML | 33G | 16G | 8G | 4G |

Table 2: Top: Averaged test accuracy over the 12 NLI tasks. Bottom: The comparison of TPU memory (High Bandwidth Memory) usage with different adaptive layer thresholds $v$.

Due to space limitation, we only report the averaged generalization error over the 12 tasks in Table 2 (top). The detailed results of the 12 NLI tasks, their standard errors, as well as the hyperparameter selections are all included in the Appendix. We also include the SOTA results from [5] for comparison and note that PACMAML is consistently the best performer over all three few-shot settings $k = 4, 8, 16$. In comparison, MAML and BMAML perform worse, possibly due to sensitivity to learning rates, as suggested by [4]. Beyond generalization errors, in Table 2 (bottom) we also compare the memory usage of MAML/BMAML against PACMAML over different adaptive layer thresholds $v$. These results emphasize the computational advantage of PACMAML by showing that as more layers are adapted (lower $v$), MAML consumes significantly more memory due to its high-order derivatives.

## 6 Discussion

We studied two PAC-Bayesian bounds for meta-learning in the few-shot case, where the number of examples in the target task is significantly smaller than that in the observed tasks. As opposed

---

[‡]Data available at: `https://github.com/iesl/leopard`.

to previous bounds, our bound in Theorem 4 remains tight in this scenario. We instantiated these new bounds and related them to the Reptile and MAML algorithms and furthermore derived the PACMAML algorithm, and showed its efficiency and effectiveness over several meta-learning benchmarks. Broadly speaking, our work falls into the category of PAC-Bayesian theories of non-i.i.d data [19]; however, our study case is more specific and our bounds are based on practical strategies. One major limitation of the work is that we do not take into account a data domain shift (e.g. [11]), which is often present in practice. However, the study of domain shift from a theoretical perspective requires additional assumptions and knowledge about the target data, which do not always exist in practice. We leave a deeper discussion and exploration on these topics to future work.

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
