# A  Proofs

In this section, we provide proofs of the main theorems presented in the paper. We also provide a brief overview of the proof of Theorem 2 from [19, 22], since the bound decomposition strategy will also be used in the new theorems of the paper.

## A.1  Brief Proof of Theorem 2 [19, 22]

Given a task environment $T$ and a set of $n$ observed tasks $(D_i, m_i) \sim T$, let $\mathcal{P}$ be a fixed hyper-prior and $\lambda > 0$, $\beta > 0$, with probability at least $1 - \delta$ over samples $S_1 \in D_1^{m_1}, \ldots, S_n \in D_n^{m_n}$, we have for all base learner $Q$ and all hyper-posterior $\mathcal{Q}$,

$$R(\mathcal{Q}, T) \leq \hat{R}(\mathcal{Q}, S_{i=1}^n) + \tilde{\xi} D_{KL}(\mathcal{Q} \| \mathcal{P})$$
$$+ \frac{1}{n\beta} \sum_{i=1}^n \mathbb{E}_{P \sim \mathcal{Q}} [D_{KL}(Q(S_i, P) \| P)] + C(\delta, \lambda, \beta, n, m_i),$$

where $\tilde{\xi} = \frac{1}{\lambda} + \frac{1}{n\beta}$.

**Proof**  The bound in Theorem 2 was proved by decomposing it into two components:

- "Task specific generalization bound", that bounds the generalization error averaged over all observed tasks $\tau_i$:

$$\mathbb{E}_{P \sim \mathcal{Q}}[\frac{1}{n} \sum_{i=1}^n L(Q(S_i, P), D_i)]$$

$$\leq \hat{R}(\mathcal{Q}, S_{i=1}^n) + \frac{1}{n\beta} D_{KL}(\mathcal{Q} \| \mathcal{P}) + \frac{1}{n\beta} \sum_{i=1}^n \mathbb{E}_{P \sim \mathcal{Q}} [D_{KL}(Q(S_i, P) \| P)]$$

$$+ \frac{1}{n\beta} \log \frac{1}{\delta} + \frac{1}{n} \sum_{i=1}^n \frac{m_i}{\beta} \Psi_1(\frac{\beta}{m_i}) \tag{15}$$

where

$$\hat{R}(\mathcal{Q}, S_{i=1}^n) = \mathbb{E}_{P \sim \mathcal{Q}}[\frac{1}{n} \sum_{i=1}^n \hat{L}(Q(S_i, P), S_i)],$$

$$\Psi_1(\beta) = \log \mathbb{E}_{P \sim \mathcal{P}} \mathbb{E}_{\mathbf{h} \sim P} \mathbb{E}_{z_{ij} \sim D_i} \left[ e^{\beta(\mathbb{E}_{z_i \sim D_i}[l(h_i, z_i)] - l(h_i, z_{ij}))} \right].$$

- "Task environment generalization bound", that bounds the transfer error from the observed tasks to the new target tasks:

$$R(\mathcal{Q}, T) \leq \frac{1}{n} \sum_{i=1}^n \mathbb{E}_{P \sim \mathcal{Q}} [L(Q(S_i, P), D_i)]$$

$$+ \frac{1}{\lambda} \left( D_{KL}(\mathcal{Q} \| \mathcal{P}) + \log \frac{1}{\delta} \right) + \frac{n}{\lambda} \Psi_2(\frac{\lambda}{n}). \tag{16}$$

where

$$\Psi_2(\lambda) = \log \mathbb{E}_{P \sim \mathcal{P}} \mathbb{E}_{D_i \sim T, S_i \sim D_i^{m_i}} \left[ e^{\lambda(\mathbb{E}_{D_i \sim T, S_i \sim D_i^{m_i}}[R_{S_i}(P)] - R_{S_i}(P))} \right].$$

Detailed proofs of these two generalization bounds can be found in the appendices of [19, 22]. Subsequently, combining Eq.(15) with Eq.(16), it is straightforward to get Eq.(4), with

$$C(\delta, \lambda, \beta, n, m_i) = \tilde{\xi} \log \frac{1}{\delta} + \frac{1}{n} \sum_{i=1}^n \frac{m_i}{\beta} \Psi_1(\frac{\beta}{m_i}) + \frac{n}{\lambda} \Psi_2(\frac{\lambda}{n}). \tag{17}$$

∎

## A.2 Proof of Theorem 3

For a target task environment $T$ and an observed task environment $\tilde{T}$ where $\mathbb{E}_{\tilde{T}}[D] = \mathbb{E}_T[D]$ and $\mathbb{E}_{\tilde{T}}[m] \geq \mathbb{E}_T[m]$, let $\mathcal{P}$ be a fixed hyper-prior and $\lambda > 0, \beta > 0$, then with probability at least $1 - \delta$ over samples $S_1 \in D_1^{m_1}, \dots, S_n \in D_n^{m_n}$ where $(D_i, m_i) \sim \tilde{T}$, we have, for all base learners $Q$ and hyper-posterior $\mathcal{Q}$,

$$R(\mathcal{Q}, T) \leq \hat{R}(\mathcal{Q}, S_{i=1}^n) + \tilde{\xi} D_{KL}(\mathcal{Q} \| \mathcal{P}) + \frac{1}{n\beta} \sum_{i=1}^n \mathbb{E}_{P \sim \mathcal{Q}} [D_{KL}(Q(S_i, P) \| P)]$$

$$+ C(\delta, \lambda, \beta, n, m_i) + \Delta_\lambda(\mathcal{P}, T, \tilde{T}),$$

where $\Delta_\lambda(\mathcal{P}, T, \tilde{T}) = \frac{1}{\lambda} \log \mathbb{E}_{P \in \mathcal{P}} e^{\lambda(R(P,T) - R(P, \tilde{T}))}$, and $\tilde{\xi} = \frac{1}{\lambda} + \frac{1}{n\beta}$.

**Proof** The "task specific generalization bound" has the same form as Eq.(15).

For the "task environment generalization bound", define the "meta-training" generalization error of a given prior $P$ on the observed task $(D_1, m_1), \dots, (D_n, m_n) \sim \tilde{T}$ as

$$R_{S_{\tilde{T}}}(P) \triangleq \frac{1}{n} \sum_{i=1}^n L(Q(S_i, P), D_i)$$

$$= \frac{1}{n} \sum_{i=1}^n \mathbb{E}_{z_i \sim D_i} \mathbb{E}_{h_i \sim Q(h_i | P, S_i)} [L(h_i, z_i)],$$

where $S_i \sim D_i^{m_i}$ and $S_{\tilde{T}} = \{S_1, \dots, S_n\}$. Similarly, the generalization error on the target task environment $T$ is

$$R(P, T) = \mathbb{E}_{(D,m) \sim T} \mathbb{E}_{S \sim D^m} \mathbb{E}_{z \in D} \mathbb{E}_{h \sim Q(h | P, S)} [L(h, z)].$$

Using the Markov Inequality, with at least $1 - \delta$ probability,

$$\mathbb{E}_{P \sim \mathcal{P}} \left[ e^{\lambda(R(P,T) - R_{S_{\tilde{T}}}(P))} \right]$$

$$\leq \frac{1}{\delta} \mathbb{E}_{P \sim \mathcal{P}} \mathbb{E}_{D_i \sim T, S_i \sim D_i^{m_i}}^{i=1,\dots,n} \left[ e^{\lambda(R(P,T) - R_{S_{\tilde{T}}}(P))} \right].$$

The left-hand side can be lower bounded by,

$$\log \mathbb{E}_{P \sim \mathcal{P}} \left[ e^{\lambda(R(P,T) - R_{S_{\tilde{T}}}(P))} \right]$$

$$= \log \mathbb{E}_{P \sim \mathcal{Q}} \frac{\mathcal{P}(P)}{\mathcal{Q}(P)} e^{\lambda(R(P,T) - R_{S_{\tilde{T}}}(P))}$$

$$\geq \mathbb{E}_{P \sim \mathcal{Q}} \log \frac{\mathcal{P}(P)}{\mathcal{Q}(P)} + \lambda \mathbb{E}_{P \sim \mathcal{Q}} [R(P,T) - R_{S_{\tilde{T}}}(P)]$$

$$= -D_{KL}(\mathcal{Q} \| \mathcal{P}) + \lambda(R(\mathcal{Q}, T) - \mathbb{E}_{P \sim \mathcal{Q}} [R_{S_{\tilde{T}}}(P)]).$$

The right-hand side is upper bounded by

$$\log \frac{1}{\delta} \mathbb{E}_{P \sim \mathcal{P}} \mathbb{E}_{D_i \sim T, S_i \sim D_i^{m_i}}^{i=1,\dots,n} \left[ e^{\lambda(R(P,T) - R_{S_{\tilde{T}}}(P))} \right]$$

$$= \log \frac{1}{\delta} + \log \mathbb{E}_{P \sim \mathcal{P}} \mathbb{E}_{D_i \sim T, S_i \sim D_i^{m_i}}^{i=1,\dots,n} \left[ e^{\lambda(R(P,T) - R_{S_{\tilde{T}}}(P))} \right]$$

$$= \log \frac{1}{\delta} + \log \mathbb{E}_{P \sim \mathcal{P}} \left[ e^{\lambda(R(P,T) - \mathbb{E}_{S_{\tilde{T}} \sim \tilde{T}} [R_{S_{\tilde{T}}}(P)])} \right]$$

$$+ \log \mathbb{E}_{P \sim \mathcal{P}} \mathbb{E}_{D_i \sim T, S_i \sim D_i^{m_i}}^{i=1,\dots,n} \left[ e^{\lambda(\mathbb{E}_{S_{\tilde{T}}} [R_{S_{\tilde{T}}}(P)] - R_{S_{\tilde{T}}}(P))} \right]$$

$$\leq \log \frac{1}{\delta} + \log \mathbb{E}_{P \sim \mathcal{P}} \left[ e^{\lambda(R(P,T) - \mathbb{E}_{S_{\tilde{T}}} [R_{S_{\tilde{T}}}(P)])} \right] + n\Psi_2(\frac{\lambda}{n}), \tag{18}$$

where,

$$
\begin{aligned}
&\mathbb{E}_{S_{\tilde{T}}}[R_{S_{\tilde{T}}}(P)] \\
&\triangleq \mathbb{E}_{(D_i,m_i)\sim\tilde{T},S_i\sim D_i^{m_i}}^{i=1,\ldots,n}[R_{S_{\tilde{T}}}(P)] \\
&= \frac{1}{n}\sum_{i=1}^{n}\mathbb{E}_{(D_i,m_i)\sim\tilde{T}}\,\mathbb{E}_{S_i\sim D_i^{m_i}}\,\mathbb{E}_{z_i\in D_i}\,\mathbb{E}_{h_i\sim Q(h_i|P,S_i)}[L(h_i,z_i)] \\
&= \mathbb{E}_{(D,m)\sim\tilde{T}}\,\mathbb{E}_{S\sim D^m}\,\mathbb{E}_{z\in D}\,\mathbb{E}_{h\sim Q(h|P,S)}[L(h,z)] \\
&= R(P,\tilde{T}).
\end{aligned}
$$

Combining the left-hand and right-hand bounds together, we have with at least probability $1-\delta$,

$$
\begin{aligned}
R(\mathcal{Q},T) \leq &\frac{1}{n}\sum_{i=1}^{n}\mathbb{E}_{P\sim\mathcal{Q}}\left[L(Q(S_i,P),D_i)\right] \\
&+ \frac{1}{\lambda}\left(D_{KL}(\mathcal{Q}\,\|\,\mathcal{P}) + \log\frac{1}{\delta} + n\Psi_2(\frac{\lambda}{n})\right) \\
&+ \frac{1}{\lambda}\log\mathbb{E}_{P\in\mathcal{P}}\,e^{\lambda(R(P,T)-R(P,\tilde{T}))}.
\end{aligned} \tag{19}
$$

Lastly, combining Eq.(19) with Eq.(15) yields Eq.(5). ∎

Furthermore, from Theorem 3, it is straightforward to obtain the following corollary.

**Corollary 5** *For a target task environment $T$ and an observed task environment $\tilde{T}$ where $\mathbb{E}_{\tilde{T}}[D] = \mathbb{E}_T[D]$ and $\mathbb{E}_{\tilde{T}}[m] \geq \mathbb{E}_T[m]$, let $\mathcal{P}$ be a fixed hyper-prior and $\lambda > 0$, $\beta > 0$, then with probability at least $1-\delta$ over samples $S_1 \in D_1^{m_1},\ldots,S_n \in D_n^{m_n}$ where $(D_i,m_i)\sim\tilde{T}$, we have, for all base learners $Q$ and hyper-posterior $\mathcal{Q}$,*

$$
\begin{aligned}
R(\mathcal{Q},T) \leq &\hat{R}(\mathcal{Q},S_{i=1}^n) + \tilde{\xi}D_{KL}(\mathcal{Q}\,\|\,\mathcal{P}) + \frac{1}{n\beta}\sum_{i=1}^{n}\mathbb{E}_{P\sim\mathcal{Q}}\left[D_{KL}(Q(S_i,P)\|P)\right] \\
&+ C(\delta,\lambda,\beta,n,m_i) + \Delta_\lambda(\mathcal{P},\mathcal{Q},T,\tilde{T}),
\end{aligned} \tag{20}
$$

*where $\Delta_\lambda(\mathcal{P},\mathcal{Q},T,\tilde{T}) = \min\left\{\frac{1}{\lambda}\log\mathbb{E}_{P\in\mathcal{P}}\,e^{\lambda(R(P,T)-R(P,\tilde{T}))}, R(\mathcal{Q},T) - R(\mathcal{Q},\tilde{T})\right\}$, and $\tilde{\xi} = \frac{1}{\lambda} + \frac{1}{n\beta}$.*

**Proof** Similar to (16), we have

$$
\begin{aligned}
R(\mathcal{Q},\tilde{T}) \leq &\frac{1}{n}\sum_{i=1}^{n}\mathbb{E}_{P\sim\mathcal{Q}}\left[L(Q(S_i,P),D_i)\right] \\
&+ \frac{1}{\lambda}\left(D_{KL}(\mathcal{Q}\,\|\,\mathcal{P}) + \log\frac{1}{\delta} + n\Psi_2(\frac{\lambda}{n})\right).
\end{aligned}
$$

A simple reorganization of the terms leads to,

$$
\begin{aligned}
R(\mathcal{Q},T) \leq &\frac{1}{n}\sum_{i=1}^{n}\mathbb{E}_{P\sim\mathcal{Q}}\left[L(Q(S_i,P),D_i)\right] \\
&+ \frac{1}{\lambda}\left(D_{KL}(\mathcal{Q}\,\|\,\mathcal{P}) + \log\frac{1}{\delta} + n\Psi_2(\frac{\lambda}{n})\right) + (R(\mathcal{Q},T) - R(\mathcal{Q},\tilde{T})).
\end{aligned} \tag{21}
$$

Combining Eq.(21) with Eq.(19) and Eq.(15) gives the bound in Eq.(20). ∎

Note that although Eq.(20) gives a potentially tighter bound than Eq.(5), empirically it makes little difference because $R(\mathcal{Q},T) - R(\mathcal{Q},\tilde{T})$ is inestimable in practice and cannot be directly optimized as a function of $\mathcal{Q}$. We will only numerically estimate its value in synthetic datasets in order to estimate the bound.

## A.3 Proof of Theorem 4

For a target task environment $T$ and an observed task environment $\tilde{T}$ where $\mathbb{E}_{\tilde{T}}[D] = \mathbb{E}_T[D]$ and $\mathbb{E}_{\tilde{T}}[m] \geq \mathbb{E}_T[m]$, let $\mathcal{P}$ be a fixed hyper-prior and $\lambda > 0$, $\beta > 0$, then with probability at least $1 - \delta$ over samples $S_1 \in D_1^{m_1}, \ldots, S_n \in D_n^{m_n}$ where $(D_i, m_i) \sim \tilde{T}$, and subsamples $S_1' \in D_1^{m_1'} \subset S_1, \ldots, S_n' \in D_n^{m_n'} \subset S_n$, where $\mathbb{E}[m_i'] = \mathbb{E}_T[m]$, we have, for all base learner $Q$ and all hyper-posterior $\mathcal{Q}$,

$$R(\mathcal{Q}, T) \leq \mathbb{E}_{P \sim \mathcal{Q}}\left[\frac{1}{n}\sum_{i=1}^{n}\hat{L}(Q(S_i', P), S_i)\right] + \tilde{\xi}D_{KL}(\mathcal{Q} \,\|\, \mathcal{P}) + \frac{1}{n\beta}\sum_{i=1}^{n}\mathbb{E}_{P \sim \mathcal{Q}}\left[D_{KL}(Q(S_i', P)\|P)\right]$$
$$+ C(\delta, \lambda, \beta, n, m_i),$$

where $\tilde{\xi} = \frac{1}{\lambda} + \frac{1}{n\beta}$.

**Proof** The "task environment generalization bound" is the same as the one in Theorem 2, because the base-learner in observed and target task have the same task environment $T$. Therefore, we have

$$R(\mathcal{Q}, T) \leq \frac{1}{n}\sum_{i=1}^{n}\mathbb{E}_{P \sim \mathcal{Q}}\left[L(Q(S_i', P), D_i)\right] + \frac{1}{\lambda}\left(D_{KL}(\mathcal{Q} \,\|\, \mathcal{P}) + \log\frac{1}{\delta}\right) + \frac{n}{\lambda}\Psi_2(\frac{\lambda}{n}). \quad (22)$$

As for the "task-specific generalization bound", define,

$$\hat{L}(\mathbf{h}) = \frac{1}{n}\sum_{i=1}^{n}\frac{1}{m_i}\sum_{j=1}^{m_i}l(h_i, z_{ij}), \quad L(\mathbf{h}) = \frac{1}{n}\sum_{i=1}^{n}\mathbb{E}_{z_i \sim D_i}\,l(h_i, z_i),$$

where $z_{ij} \in S_i$ which is sampled from $D_i$. According to the Markov inequality, with at least $1 - \delta$ probability, we have

$$\mathbb{E}_{P \sim \mathcal{P}}\,\mathbb{E}_{\mathbf{h} \sim P^n}\left[e^{n\beta(L(\mathbf{h})-\hat{L}(\mathbf{h}))}\right] \leq \frac{1}{\delta}\mathbb{E}_{P \sim \mathcal{P}}\,\mathbb{E}_{\mathbf{h} \sim P^n}\,\mathbb{E}_{\mathbf{S} \sim \mathbf{D^m}}\left[e^{n\beta(L(\mathbf{h})-\hat{L}(\mathbf{h}))}\right]$$

Now take the logarithm of both sides, and transform the expectation over $\mathcal{P}, P$ to $\mathcal{Q}, Q$, where we use base-learner $Q(S_i', P)$ with $S_i' \in D_i^{m_i'}$. Then the LHS becomes

$$\log \mathbb{E}_{P \sim \mathcal{P}}\,\mathbb{E}_{\mathbf{h} \sim P^n}\left[e^{n\beta(L(\mathbf{h})-\hat{L}(\mathbf{h}))}\right]$$
$$= \log \mathbb{E}_{P \sim \mathcal{Q}}\,\mathbb{E}_{\mathbf{h} \sim \mathbf{Q}(\mathbf{S'},P)}\left[\frac{\mathcal{P}(P)\prod_{i=1}^{n}P(h_i)}{\mathcal{Q}(P)\prod_{i=1}^{n}Q_i(h_i|S_i', P)}e^{n\beta(L(\mathbf{h})-\hat{L}(\mathbf{h}))}\right]$$
$$\geq -D_{KL}(\mathcal{Q} \,\|\, \mathcal{P}) - \sum_{i=1}^{n}\mathbb{E}_{P \sim \mathcal{Q}}\left[D_{KL}(Q(S_i', P)\|P)\right]$$
$$+ \beta\,\mathbb{E}_{P \sim \mathcal{Q}}[\sum_{i=1}^{n}L(Q(S_i', P), D_i)] - \beta\,\mathbb{E}_{P \sim \mathcal{Q}}[\sum_{i=1}^{n}\hat{L}(Q(S_i', P), S_i)].$$

The first equation uses the fact that the hyper-prior $\mathcal{P}$ and hyper-posterior $\mathcal{Q}$ as well as the prior $P$ are shared across all $n$ observed tasks. The inequality uses Jensen's inequality to move the logarithm inside expectation.

The RHS is

$$\log\frac{1}{\delta} + \log \mathbb{E}_{P \sim \mathcal{P}}\,\mathbb{E}_{\mathbf{h} \sim P^n}\,\mathbb{E}_{\mathbf{S} \sim \mathbf{D^m}}\left[e^{n\beta(L(h)-\hat{L}(h))}\right]$$
$$= \log\frac{1}{\delta} + \log \mathbb{E}_{P \sim \mathcal{P}}\,\mathbb{E}_{\mathbf{h} \sim P^n}\prod_{i=1}^{n}\prod_{j=1}^{m_i}\mathbb{E}_{z_{ij} \sim D_i}\left[e^{\frac{\beta}{m_i}(\mathbb{E}_{z_i \sim D_i}[l(h_i, z_i)]-l(h_i, z_{ij}))}\right]$$
$$= \log\frac{1}{\delta} + \sum_{i=1}^{n}m_i\Psi_1(\frac{\beta}{m_i}).$$

Now, combining the LHS and RHS together, we get that with at least $1 - \delta$ probability,

$$\mathbb{E}_{P \sim \mathcal{Q}}[\frac{1}{n} \sum_{i=1}^{n} L(Q(S_i', P), D_i)]$$

$$\leq \mathbb{E}_{P \sim \mathcal{Q}} \left[ \frac{1}{n} \sum_{i=1}^{n} L(Q(S_i', P), S_i) \right] + \frac{1}{n\beta} D_{KL}(\mathcal{Q} \, \| \, \mathcal{P}) + \frac{1}{n\beta} \sum_{i=1}^{n} \mathbb{E}_{P \sim \mathcal{Q}} \left[ D_{KL}(Q(S_i', P) \| P) \right]$$

$$+ \frac{1}{n\beta} \log \frac{1}{\delta} + \frac{1}{n} \sum_{i=1}^{n} \frac{m_i}{\beta} \Psi_1(\frac{\beta}{m_i}). \tag{23}$$

Combining Eq.(22) with Eq.(23) immediately yields Eq.(6). ∎

## B Derivations of MAML and Reptile

In this section, we derive a couple of meta-learning algorithms based on the MAP estimation of PAC-Bayesian bounds. To this end, we assume that the distribution families of the hyper-posterior $\mathcal{Q}(P)$ and posterior $Q_i(h)$ are from delta functions. In addition, we use the isotrophic Gaussian priors for the hyper-prior $\mathcal{P}(P)$ and the prior $P(h)$ on all model parameters,

$$\mathcal{P}(P) = \mathcal{N}(\mathbf{p} \, | 0, \sigma_0^2)$$
$$\mathcal{Q}(P) = \delta(\mathbf{p} = \mathbf{p}_0)$$
$$P(h) = \mathcal{N}(\mathbf{h} \, | \, \mathbf{p}, \sigma^2)$$
$$Q_i(h) = \delta(\mathbf{h} = \mathbf{q}_i) \; \forall i = 1, \dots, n.$$

This way we have a closed form solution for the two KL terms, which are (up to a constant)

$$D_{KL}(\mathcal{Q} \, \| \, \mathcal{P}) = \int d\mathbf{p} \, \delta(\mathbf{p} = \mathbf{p}_0) \cdot \left( \frac{\|\mathbf{p}\|^2}{2\sigma_0^2} + \frac{k}{2} \log(2\pi\sigma_0^2) + \log \delta(\mathbf{p} = \mathbf{p}_0) \right)$$

$$= \frac{\|\mathbf{p}_0\|^2}{2\sigma_0^2} + \frac{k}{2} \log(2\pi\sigma_0^2) + c,$$

where $k$ is the dimension of $\mathbf{p}$ and $c$ is a constant. Similarly,

$$\mathbb{E}_{P \sim \mathcal{Q}}[D_{KL}(Q_i | P)]$$

$$= \int d\mathbf{p} \, \delta(\mathbf{p} = \mathbf{p}_0) \int d\mathbf{h} \, \delta(\mathbf{h} = \mathbf{q}_i) \cdot \left( \frac{\|\mathbf{h} - \mathbf{p}\|^2}{2\sigma^2} + \frac{k}{2} \log(2\pi\sigma^2) + \log \delta(\mathbf{h} = \mathbf{q}_i) \right)$$

$$= \int d\mathbf{p} \, \delta(\mathbf{p} = \mathbf{p}_0) \cdot \frac{\|\mathbf{p} - \mathbf{q}_i\|^2}{2\sigma^2} + \frac{k}{2} \log(2\pi\sigma^2) + c$$

$$= \frac{\|\mathbf{p}_0 - \mathbf{q}_i\|^2}{2\sigma^2} + \frac{k}{2} \log(2\pi\sigma^2) + c.$$

Plugging in the above results, the PAC-Bayesian bound ($PacB$) in Eq.(5) and Eq.(6) are both of the form of,

$$PacB = \frac{1}{n} \sum_{i=1}^{n} L(\mathbf{q}_i, S_i) + \frac{\tilde{\xi} \|\mathbf{p}_0\|^2}{2\sigma_0^2} + \frac{1}{n\beta} \sum_{i=1}^{n} \frac{\|\mathbf{p}_0 - \mathbf{q}_i\|^2}{2\sigma^2} + C',$$

where the constant $C'$ corresponding to Eq.(5) and Eq.(6) are different by $\Delta_\lambda$. The only free variable of $PacB$ is $\mathbf{p}_0$. The base-learner $\mathbf{q}_i$ can be any function of $\mathbf{p}_0$ and $S_i$ for Eq.(5) or $S_i'$ for Eq.(6). One could find the MAP estimation of $PacB$ by gradient descent with respect to $\mathbf{p}_0$.

Note that in Eq.(5), for a given $\mathbf{p}_0$ and $S_i$, there exists an optimal base-learner $\mathbf{q}_i^*$ in the form of,

$$\mathbf{q}_i^* = \underset{\mathbf{q}_i}{\operatorname{argmin}}(PacB) = \underset{\mathbf{q}_i}{\operatorname{argmin}} \left[ L(\mathbf{q}_i, S_i) + \frac{\|\mathbf{p}_0 - \mathbf{q}_i\|^2}{2\beta\sigma^2} \right].$$

Given the optimal $\mathbf{q}_i^*$, the full derivative of $PacB$ with respect to $\mathbf{p}_0$ is substantially simpler,

$$\frac{d(PacB)}{d\mathbf{p}_0} = \frac{\partial(PacB)}{\partial\mathbf{p}_0} + \left\langle \frac{\partial\mathbf{q}_i^*}{\partial\mathbf{p}_0}, \frac{\partial(PacB)}{\partial\mathbf{q}_i^*} \right\rangle$$

$$= \frac{\partial(PacB)}{\partial\mathbf{p}_0} = \frac{\tilde{\xi}\,\mathbf{p}_0}{\sigma_0^2} + \frac{1}{n}\sum_{i=1}^{n}\frac{\mathbf{p}_0 - \mathbf{q}_i^*}{\beta\sigma^2}, \qquad (24)$$

where the 2nd equation is because $\frac{\partial(PacB)}{\partial\mathbf{q}_i^*} = 0$ for the optimal base-leaner $\mathbf{q}_i^*$. Eq.(24) is the equivalent to the meta-update of the Reptile algorithm [16], except that Reptile does not solve for the optimal base learner $\mathbf{q}_i^*$.

From the optimal condition, the base-learner $\mathbf{q}_i^*$ satisfies,

$$\frac{\mathbf{p}_0 - \mathbf{q}_i^*}{\beta\sigma^2} = \nabla_{\mathbf{q}_i^*}L(\mathbf{q}_i^*, S_i).$$

Therefore, we can rewrite Eq.(24) in the form of the *implicit gradient*,

$$\frac{d(PacB)}{d\mathbf{p}_0} = \frac{\tilde{\xi}\,\mathbf{p}_0}{\sigma_0^2} + \frac{1}{n}\sum_{i=1}^{n}\nabla_{\mathbf{q}_i^*}L(\mathbf{q}_i^*, S_i).$$

In contrast, the standard multi-task objective uses the *explicit gradient*, where $\mathbf{q}_i = \mathbf{p}_0$ and

$$\frac{d(PacB)}{d\mathbf{p}_0} = \frac{\tilde{\xi}\,\mathbf{p}_0}{\sigma_0^2} + \frac{1}{n}\sum_{i=1}^{n}\nabla_{\mathbf{p}_0}L(\mathbf{p}_0, S_i).$$

## C  Derivations of PACMAML

For Theorem 4, we use the following posterior as the base-learner for observed task $\tau_i$,

$$Q_i(S_i', P)(h) = \frac{P(h)\exp(-\alpha\hat{L}(h, S_i'))}{Z_\alpha(S_i', P)}.$$

Plugging this $Q_i$ into Eq.(6), we have

$$R(\mathcal{Q}, T)$$

$$\leq \mathbb{E}_{P\sim\mathcal{Q}}\left[\frac{1}{n}\sum_{i=1}^{n}\hat{L}(Q_i, S_i)\right] + \tilde{\xi}D_{KL}(\mathcal{Q}\,\|\,\mathcal{P}) + \frac{1}{n\beta}\sum_{i=1}^{n}\mathbb{E}_{P\sim\mathcal{Q}}\left[D_{KL}(Q_i\|P)\right] + C$$

$$= \frac{1}{n}\sum_{i=1}^{n}\mathbb{E}_{P\sim\mathcal{Q}}\left[\hat{L}(Q_i, S_i) + \frac{1}{\beta}D_{KL}(Q_i\|P)\right] + \tilde{\xi}D_{KL}(\mathcal{Q}\,\|\,\mathcal{P}) + C$$

$$= \frac{1}{n}\sum_{i=1}^{n}\mathbb{E}_{P\sim\mathcal{Q}}\,\mathbb{E}_{h\sim Q_i}\left[\hat{L}(h, S_i) + \frac{1}{\beta}\log Q_i(h) - \frac{1}{\beta}\log P(h)\right] + \tilde{\xi}D_{KL}(\mathcal{Q}\,\|\,\mathcal{P}) + C$$

$$= \frac{1}{n}\sum_{i=1}^{n}\mathbb{E}_{P\sim\mathcal{Q}}\,\mathbb{E}_{h\sim Q_i}\left[\hat{L}(h, S_i) - \frac{\alpha}{\beta}\hat{L}(h, S_i') - \frac{1}{\beta}\log Z_\alpha(S_i', P))\right] + \tilde{\xi}D_{KL}(\mathcal{Q}\,\|\,\mathcal{P}) + C$$

$$= \frac{1}{n}\sum_{i=1}^{n}\mathbb{E}_{P\sim\mathcal{Q}}[-\frac{1}{\beta}\log Z_\alpha(S_i', P) + \hat{L}(Q_i, S_i) - \frac{\alpha}{\beta}\hat{L}(Q_i, S_i')] + \tilde{\xi}D_{KL}(\mathcal{Q}\,\|\,\mathcal{P}) + C.$$

where $C = \tilde{\xi}\log\frac{2}{\delta} + \frac{n}{\lambda}\Psi(\frac{\lambda}{n}) + \frac{1}{n}\sum_{i=1}^{n}\frac{m_i}{\beta}\Psi(\frac{\beta}{m_i})$.

### C.1  The Gradient Estimator of PACOH and PACMAML

Assuming that the model hypothesis $h$ is parameterized by $\mathbf{v}$ such that $\hat{L}(h, S_i) \triangleq \hat{L}(\mathbf{v}, S_i)$, and $\mathbf{v}$ has prior $P(\mathbf{v}) = \mathcal{N}(\mathbf{v}\,|\,\mathbf{p}, \sigma^2)$ with meta-parameter $\mathbf{p}$, then

$$\log Z_\beta(S_i, \mathbf{p}) = \log\int\mathcal{N}(\mathbf{v}\,|\,\mathbf{p}, \sigma^2)\exp(-\beta\hat{L}(\mathbf{v}, S_i))d\mathbf{v}.$$

Note that the parameter $\mathbf{p}$ appears in the probability distribution of the expectation, and the naive Monte-Carlo gradient estimator of such gradient is known to exhibit high variance. To reduce the variance, we apply the reparameterization trick [13] and rewrite $\mathbf{v} = \mathbf{p} + \mathbf{w}$ with $\mathbf{w} \sim \mathcal{N}(\mathbf{w} \mid \mathbf{0}, \sigma^2)$, then

$$\log Z_\beta(S_i, \mathbf{p}) = \log \int \mathcal{N}(\mathbf{w} \mid \mathbf{0}, \sigma^2) \exp(-\beta \hat{L}(\mathbf{p} + \mathbf{w}, S_i)) d\mathbf{w}.$$

This leads to the gradient of $W_1$ in the following form,

$$\frac{d}{d\mathbf{p}} W_1 = -\frac{1}{\beta} \frac{d}{d\mathbf{p}} \log Z_\beta(S_i, \mathbf{p}) = \int Q_i^\beta(\mathbf{w}; S_i) \frac{\partial \hat{L}(\mathbf{p} + \mathbf{w}, S_i)}{\partial \mathbf{p}} d\mathbf{w},$$

$$\text{where, } Q_i^\beta(\mathbf{w}; S_i) \propto \mathcal{N}(\mathbf{w} \mid \mathbf{0}, \sigma^2) \exp(-\beta \hat{L}(\mathbf{p} + \mathbf{w}, S_i)).$$

As for $W_2$, the first term is similar to $W_1$, but we also need to evaluate the gradient of $\hat{L}_{\frac{\alpha}{\beta}}^{\Delta}(Q_i^\alpha, S_i, S_i')$, which is

$$\frac{d}{d\mathbf{p}} \hat{L}_{\frac{\alpha}{\beta}}^{\Delta}(Q_i^\alpha, S_i, S_i') = \int Q_i^\alpha(\mathbf{w}; S_i') \frac{\partial \hat{L}_{\frac{\alpha}{\beta}}^{\Delta}(\mathbf{p} + \mathbf{w}, S_i, S_i')}{\partial \mathbf{p}} d\mathbf{w} + \int \frac{\partial Q_i^\alpha(\mathbf{w}; S_i')}{\partial \mathbf{p}} \hat{L}_{\frac{\alpha}{\beta}}^{\Delta}(\mathbf{p} + \mathbf{w}, S_i, S_i') d\mathbf{w}.$$
$$(25)$$

The second term of Eq.(25) is equivalent to,

$$\int \frac{\partial Q_i^\alpha(\mathbf{w}; S_i')}{\partial \mathbf{p}} \hat{L}_{\frac{\alpha}{\beta}}^{\Delta}(\mathbf{p} + \mathbf{w}, S_i, S_i') d\mathbf{w}$$
$$= -\frac{1}{\beta} \frac{\partial}{\partial \mathbf{p}} \int Q_i^\alpha(\mathbf{w}; S_i') \text{stop\_grad}\left(-\beta \hat{L}_{\frac{\alpha}{\beta}}^{\Delta}(\mathbf{p} + \mathbf{w}, S_i, S_i')\right) d\mathbf{w}.$$

The Monte-Carlo gradient estimator of this has the same high-variance problem as in the policy gradient method, which causes unreliable inference without warm-start. Instead, we apply the cold-start policy gradient method by approximating the loss with the one from the softmax value function [8] as follows,

$$-\frac{1}{\beta} \int Q_i^\alpha(\mathbf{w}; S_i') \text{stop\_grad}\left(-\beta \hat{L}_{\frac{\alpha}{\beta}}^{\Delta}(\mathbf{p} + \mathbf{w}, S_i, S_i')\right) d\mathbf{w}$$
$$\geq -\frac{1}{\beta} \log \int Q_i^\alpha(\mathbf{w}; S_i') \exp\left(\text{stop\_grad}\left(-\beta \hat{L}_{\frac{\alpha}{\beta}}^{\Delta}(\mathbf{p} + \mathbf{w}, S_i, S_i')\right)\right) d\mathbf{w}.$$

Then we take the gradient of the softmax value function,

$$-\frac{1}{\beta} \frac{\partial}{\partial \mathbf{p}} \log \int Q_i^\alpha(\mathbf{w}; S_i') \exp\left(\text{stop\_grad}\left(-\beta \hat{L}_{\frac{\alpha}{\beta}}^{\Delta}(\mathbf{p} + \mathbf{w}, S_i, S_i')\right)\right) d\mathbf{w}$$
$$= -\frac{1}{\beta} \frac{\int \frac{\partial Q_i^\alpha(\mathbf{w}; S_i')}{\partial \mathbf{p}} \exp\left(\text{stop\_grad}\left(-\beta \hat{L}_{\frac{\alpha}{\beta}}^{\Delta}(\mathbf{p} + \mathbf{w}, S_i, S_i')\right)\right) d\mathbf{w}}{\int Q_i^\alpha(\mathbf{w}; S_i') \exp\left(\text{stop\_grad}\left(-\beta \hat{L}_{\frac{\alpha}{\beta}}^{\Delta}(\mathbf{p} + \mathbf{w}, S_i, S_i')\right)\right) d\mathbf{w}}$$
$$= -\frac{1}{\beta} \frac{\int \frac{\partial \log Q_i^\alpha(\mathbf{w}; S_i')}{\partial \mathbf{p}} Q_i^\alpha(\mathbf{w}; S_i') \exp\left(\text{stop\_grad}\left(-\beta \hat{L}_{\frac{\alpha}{\beta}}^{\Delta}(\mathbf{p} + \mathbf{w}, S_i, S_i')\right)\right) d\mathbf{w}}{\int Q_i^\alpha(\mathbf{w}; S_i') \exp\left(\text{stop\_grad}\left(-\beta \hat{L}_{\frac{\alpha}{\beta}}^{\Delta}(\mathbf{p} + \mathbf{w}, S_i, S_i')\right)\right) d\mathbf{w}}$$
$$= -\frac{1}{\beta} \frac{\int \frac{\partial \log Q_i^\alpha(\mathbf{w}; S_i')}{\partial \mathbf{p}} \mathcal{N}(\mathbf{w} \mid \mathbf{0}, \sigma^2) \exp(-\beta \hat{L}(\mathbf{p} + \mathbf{w}, S_i)) d\mathbf{w}}{\int \mathcal{N}(\mathbf{w} \mid \mathbf{0}, \sigma^2) \exp(-\beta \hat{L}(\mathbf{p} + \mathbf{w}, S_i)) d\mathbf{w}}$$
$$= -\frac{1}{\beta} \int Q_i^\beta(\mathbf{w}; S_i) \frac{\partial \log Q_i^\alpha(\mathbf{w}; S_i')}{\partial \mathbf{p}} d\mathbf{w}$$
$$= \frac{\alpha}{\beta} \int \left(Q_i^\beta(\mathbf{w}; S_i) - Q_i^\alpha(\mathbf{w}; S_i')\right) \frac{\partial \hat{L}(\mathbf{p} + \mathbf{w}, S_i')}{\partial \mathbf{p}} d\mathbf{w}.$$

This yields the overall gradient of $W_2$ to be,

$$\frac{d}{d\mathbf{p}}W_2 \simeq \frac{\alpha}{\beta}\int Q_i^{\alpha}(\mathbf{w};S_i')\frac{\partial \hat{L}(\mathbf{p}+\mathbf{w},S_i')}{\partial \mathbf{p}}d\mathbf{w} + \int Q_i^{\alpha}(\mathbf{w};S_i')\frac{\partial \hat{L}_{\frac{\alpha}{\beta}}^{\Delta}(\mathbf{p}+\mathbf{w},S_i,S_i')}{\partial \mathbf{p}}d\mathbf{w}$$

$$+\frac{\alpha}{\beta}\int \left(Q_i^{\beta}(\mathbf{w};S_i) - Q_i^{\alpha}(\mathbf{w};S_i')\right)\frac{\partial \hat{L}(\mathbf{p}+\mathbf{w},S_i')}{\partial \mathbf{p}}d\mathbf{w}$$

$$=\frac{\alpha}{\beta}\int Q_i^{\beta}(\mathbf{w};S_i)\frac{\partial \hat{L}(\mathbf{p}+\mathbf{w};S_i')}{\partial \mathbf{p}}d\mathbf{w} + \int Q_i^{\alpha}(\mathbf{w};S_i')\frac{\partial \hat{L}_{\frac{\alpha}{\beta}}^{\Delta}(\mathbf{p}+\mathbf{w},S_i,S_i')}{\partial \mathbf{p}}d\mathbf{w}$$

$$=\int Q_i^{\alpha}(\mathbf{w};S_i')\frac{\partial \hat{L}(\mathbf{p}+\mathbf{w};S_i)}{\partial \mathbf{p}}d\mathbf{w} + \frac{\alpha}{\beta}\int \left(Q_i^{\beta}(\mathbf{w};S_i) - Q_i^{\alpha}(\mathbf{w};S_i')\right)\frac{\partial \hat{L}(\mathbf{p}+\mathbf{w};S_i')}{\partial \mathbf{p}}d\mathbf{w}.$$

The Pseudocode of PACMAML is shown in Algorithm 1.

---

**Algorithm 1** Pseudocode of PACMAML with approximate gradient estimation. Every posterior is approximated by 1 sample of SVGD, which reduces to SGD. For notation simplicity, we also assume both inner and outer loop uses a gradient decent with fixed learning rate.

---

Input: $\sigma$, $\eta$, $\lambda$, $\alpha$, $\beta$, $N$, $K$.
Initialize: $\mathbf{p}_0$.
**for** $i = 0, \ldots, N-1$ **do**
    $\mathbf{w}_{i,0}^{\alpha} = 0, \mathbf{w}_{i,0}^{\beta} = 0$
    **for** $k = 0, \ldots, K-1$ **do**
        $\mathbf{w}_{i,k+1}^{\alpha} = \mathbf{w}_{i,k}^{\alpha} - \eta\left(\log\mathcal{N}(\mathbf{w}_{i,k}\,|0,\sigma^2) - \beta\hat{L}(\mathbf{p}_i + \mathbf{w}_{i,k}, S_i')\right)$
        $\mathbf{w}_{i,k+1}^{\beta} = \mathbf{w}_{i,k}^{\beta} - \eta\left(\log\mathcal{N}(\mathbf{w}_{i,k}\,|0,\sigma^2) - \alpha\hat{L}(\mathbf{p}_i + \mathbf{w}_{i,k}, S_i)\right)$
    **end for**
    $\mathbf{p}_{i+1} = \mathbf{p}_i - \lambda\nabla_p\left(\hat{L}(\mathbf{p}_i + \mathbf{w}_{i,K}^{\alpha}, S_i) - \frac{\alpha}{\beta}\hat{L}(\mathbf{p}_i + \mathbf{w}_{i,K}^{\alpha}, S_i') + \frac{\alpha}{\beta}\hat{L}(\mathbf{p}_i + \mathbf{w}_{i,K}^{\beta}, S_i)\right)$
**end for**
Output: $\mathbf{p}_N$.

---

# D  Experiment Details of the Regression Problem

## D.1  Gaussian Process Model Details

We use the Gaussian process prior, where $P_\theta(h) = \mathcal{GP}(h|m_\theta(x), k_\theta(x, x'))$ and $k_\theta(x, x') = \frac{1}{2}\exp\left(-\|\phi_\theta(x) - \phi_\theta(x')\|^2\right)$. Both $m_\theta(x)$ and $\phi_\theta(x)$ are instantiated to be neural networks. The networks are composed of an input layer of size $1 \times 32$, a hidden layer of size $32 \times 32$. $m_\theta$ and $\phi_\theta$ has an output layer of size $32 \times 1$ and $32 \times 2$, respectively.

We focused on regression problems where for every example $z_j = (x_j, y_j)$ and a hypothesis $h$, the $l_2$-loss function is used so that $l(h, z_j) = \|h(x_j) - y_j\|_2^2$. This leads to a Gaussian likelihood function. Assuming there are $m$ examples in the dataset, we have

$$P(y|h, x) = \mathcal{N}(h, \frac{m}{2\alpha}I)$$

$$= \frac{1}{(\pi m/\alpha)^{m/2}}\exp\left(-\frac{\alpha}{m}\sum_{j=1}^{m}(h(x_j) - y_j)^2\right).$$

As a result, the partition function $Z_\alpha(S, P)$ is,

$$Z_\alpha(S, P) = (\pi m/\alpha)^{m/2}\int_h dh P(y|h, x)P_\theta(h)$$

$$= (\pi m/\alpha)^{m/2}\mathcal{N}(y|m_\theta(x), k_\theta(x, x') + \frac{m}{2\alpha}I),$$

We apply the GP base-learner $Q$ on the the observed data $S_i$ of task $\tau_i$. For notation simplicity, let us denote $Q_i(h^i|S_i, P) = \mathcal{N}(\mu_i, K_i)$, where $h^i$ denotes the model hypothesis (predictions) of the $m_i$ examples in $S_i$. Then we have,

$$
\begin{aligned}
\hat{L}(Q_i, S_i) =& \frac{1}{m_i} \int Q_i(h^i)(y^i - h^i)^\top (y^i - h^i) dh^i \\
=& \frac{1}{m_i} \left( y^{i\top} y^i - 2\mu_i^\top y^i + \mu_i^\top \mu_i + \mathrm{tr}(K_i) \right),
\end{aligned}
$$

where $y^i$ denotes the labels of the $m_i$ examples in $S_i$.

The hyper-prior $\mathcal{P}(P_\theta) := \mathcal{P}(\theta) = \mathcal{N}(\theta|0, \sigma_0^2 I)$ is an isotropic Gaussian defined over the network parameters $\theta$, where we take $\sigma_0^2 = 3$ in our numerical experiments. The MAP approximated hyper-posterior takes the form of a delta function, where $\mathcal{Q}_{\theta_0}(P_\theta) := \mathcal{Q}_{\theta_0}(\theta) = \delta(\theta = \theta_0)$. As a result, we have

$$
\begin{aligned}
& D_{KL}(\mathcal{Q}_{\theta_0} \parallel \mathcal{P}) \\
=& \int d\theta \delta(\theta = \theta_0) \left( \frac{\|\theta\|^2}{2\sigma_0^2} + \frac{k}{2} \log(2\pi\sigma_0^2) + \log \delta(\theta = \theta_0) \right) \\
=& \frac{\|\theta_0\|^2}{2\sigma_0^2} + \frac{k}{2} \log(2\pi\sigma_0^2) + c,
\end{aligned}
$$

which combined with $\tilde{\xi}$ becomes the regularizer on the parameters $\theta_0$.

### D.2 Experiment Details

In the Sinusoid experiment, the number of available examples per observed task $m_i \in \{5, 10, 30, 50, 100\}$. Under the setting of PACOH (Theorem 3), for each different $m_i$, we did a grid search on $\beta/m_i \in \{10, 30, 100\}$. Under the setting of PACMAML (Theorem 4), for each different $m_i$, we did a grid search on $\beta/m_i \in \{10, 30, 100\}$ and $\alpha/\beta \in \{0.1, 0.2, 0.3, 0.4, 0.5, 0.6\}$. We use a subsect $S_i' \subset S_i$ with $m_i' = m$ to train the base-learner in PACMAML. For each hyperparameter setting $\beta$ (and $\alpha$), we trained 40 models. Each model is trained on 1 of the 8 pre-sampled meta-training sets (each containing $n = 20$ observed tasks) and each set is run with 5 random seeds of network initialization. The ultimate result for each $\beta$ (and $\alpha$) is the averaged result across all models of that setting. The hyperparameters $\tilde{\xi}$ and $\sigma_0^2$ in the hyper-prior ($\mathcal{P}(\theta) = \mathcal{N}(\theta|0, \sigma_0^2 I)$) are chosen to be $\tilde{\xi} = 1/(n\beta)$ and $\sigma_0^2 = 3$. To find the optimal model parameter $\theta_0$, we used the ADAM optimizer with learning rate $3 \times 10^{-3}$. The number of tasks per batch is fixed to 5 across all experiments. We run 8000 iterations for each experiment.

The experiments ran in parallel on several 56-core Intel CLX processors and each experiment runs on a single core. Each iteration in the PACOH and PACMAML setting takes about 0.03-0.06s and 0.07-0.14s to run, respectively, with the exact run-time varying for different number of tasks $n$ and number of examples $m_i$.

### D.3 Additional Results

We performed the 4-fold cross validation over the 20 target tasks to determine the optimal $\beta$ for PACOH (Theorem 3) or the optimal $\alpha$ and $\beta$ for PACMAML (Theorem 4). For the selected $\alpha$ and $\beta$ form validation, we report the lowest test error the corresponding models can achieve. The results are plotted in Figure 3. For each setting, both the validation and test errors show the same trend, where the error with PACOH setting saturates earlier than that with PACMAML setting.

| $m_i$ | 5 | 10 | 30 | 50 | 100 |
|---|---|---|---|---|---|
| $\beta/m_i$ | 100 | 100 | 30 | 30 | 100 |

Table 3: Optimal $\beta$ under the setting of PACOH, based on the results of a 4-fold cross validation.

In Table 3 and Table 4, we provide the optimal $\beta$ (and $\alpha$) for PACOH and PACMAML, respectively. In Fig. 4, we plotted the validation error for three different values of $\beta$ we used. We see that for both PACOH and PACMAML, the error is large for a small $\beta/m_i = 10$. The error with $\beta/m_i = 30$ and

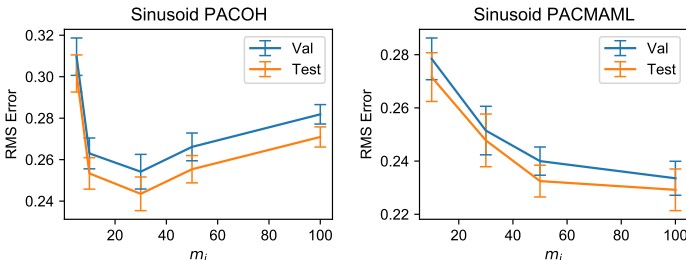

Figure 3: The validation and test error (error bars corresponding to standard errors) on the Sinusoid dataset under the settings of PACOH and PACMAML.

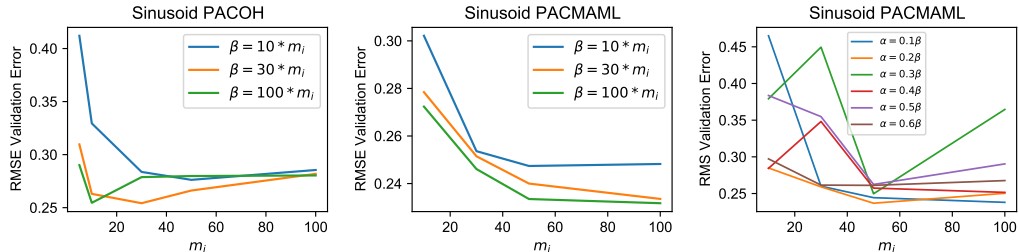

Figure 4: Left: $\beta$-dependence of the RMSE validation error under the PACOH (Theorem 3) setting. Middle and Right: $\beta$- and $\alpha$-dependence of the RMSE validation error under the PACMAML (Theorem 4) setting. $\alpha$ is chosen as the optimal $\alpha$ in the middle plot. $\beta = 30 * m_i$ in the right plot.

| $m_i$ | 10 | 30 | 50 | 100 |
|---|---|---|---|---|
| $\alpha/\beta$ | 0.2 | 0.2 | 0.2 | 0.1 |
| $\beta/m_i$ | 100 | 100 | 100 | 100 |

Table 4: Optimal $\alpha$ and $\beta$ values under the setting of PACMAML, based on the results of a 4-fold cross validation.

$\beta/m_i = 100$ are similar for PACOH. For PACMAML, the error with $\beta/m_i = 100$ is slightly and consistently better than the error with $\beta/m_i = 30$. From the right figure of Fig. 4 we see that for PACMAML, given $\beta/m_i = 30$, $\alpha/\beta$ around 0.2 achieves lowest validation error.

### D.4 Generalization Bound of PACMAML

When $\beta/m_i$ is held as a constant, the $\Psi_1$ and $\Psi_2$ terms of $C(\delta, \lambda, \beta, n, m_i)$ in Eq.(17) becomes the same across all $m_i$ and both PACOH (Eq. (10)) and PACMAML (Eq. (11)). Thus, we exclude the $\Psi_1$ and $\Psi_2$ terms when comparing the bound values for different $m_i$ and different setups PACOH and PACMAML. In Fig. 5 and 6 we show the value of each term and the total bound for PACOH and PACMAML obtained from the same set of experiments for Fig. 2-4. For both PACOH and PACMAML, all three terms $W$, $\tilde{\xi} D_{KL}$ and $\tilde{\xi} \log(1/\delta)$ tend to decrease with larger $m_i$. For PACOH, with the extra term $\Delta_\lambda$ that panalizes larger $m_i$, the total bound either always increases with $m_i$ or first increases then saturates. For PACMAML, without the $\Delta_\lambda$ term, the total bound $W_2 + \tilde{\xi} D_{KL} + \tilde{\xi} \log(1/\delta)$ monotonically decreases vs. $m_i$.

In Fig. 7, we show the comparison between the total bound of PACOH and PACMAML. We see that for all $m_i > 5$, PACMAML has lower bound for all choices of $\beta$.

### D.5 Experiment for Reptile and MAML

We also experimented with meta-learning algorithms that use Dirac-measure base-learners, by implementing the Reptile (with optimal $\mathbf{q}^*$) and the MAML algorithms following the equations of Section 3.2.

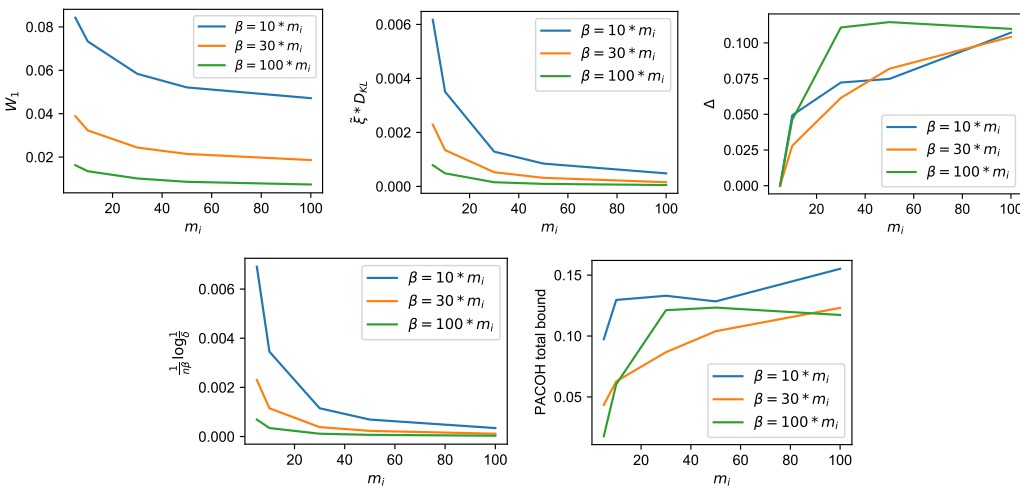

Figure 5: Values of $W_1$, $\tilde{\xi} D_{KL}$ and $\Delta$ terms in the PACOH bound and the total value of the bound for $\beta/m_i \in \{10, 30, 100\}$.

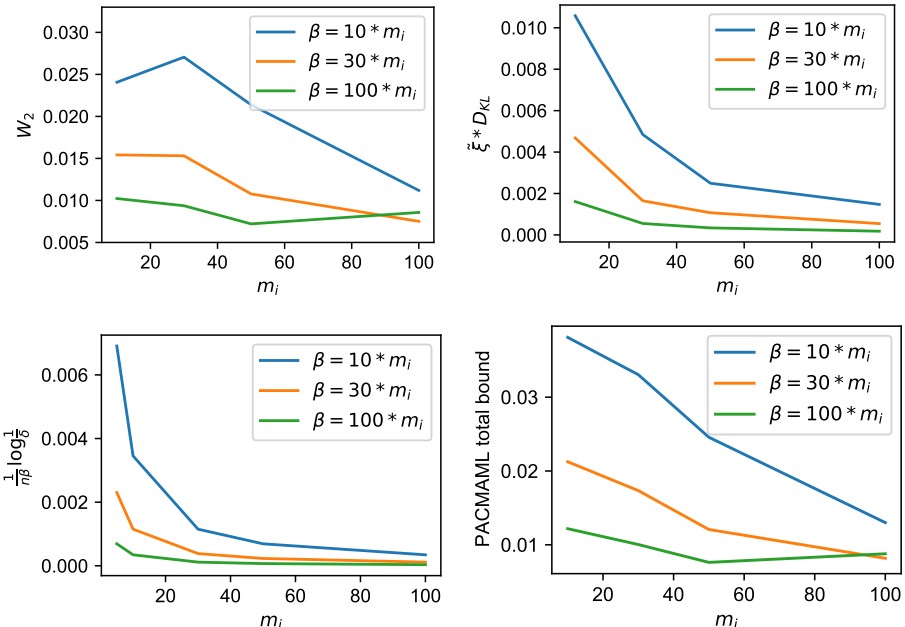

Figure 6: Values of $W_2$ and $\tilde{\xi} D_{KL}$ terms in the PACMAML bound and the total value of the bound for $\beta/m_i \in \{10, 30, 100\}$. $\alpha$ for each $m_i$ is set to the optimal value according to Fig. 4.

Reptile follows the same experiment setting as PACOH. MAML follows the same experiment setting as PACMAML where $S_i' \subset S_i$, $m_i' = m$. In order to compute the optimal $\mathbf{q}_i^*$ for Reptile, we use an L-BFGS optimizer in the inner loop with `lr = 5e-3, history_size = 10, max_iter =10`. Other experiment setting and hyperparameter selection procedure are the same as those in Section D.3.

The results of the 4-fold cross validation are plotted in Fig. 8. The errors of Reptile and MAML follow a very similar trend to the ones with non-Dirac measure base-learners under PACOH and PACMAML setting, respectively (Fig. 3). However, the models with non-Dirac measure base-learners appear to have lower generalization errors than the ones with Dirac measure base-learners (i.e. Reptile and MAML).

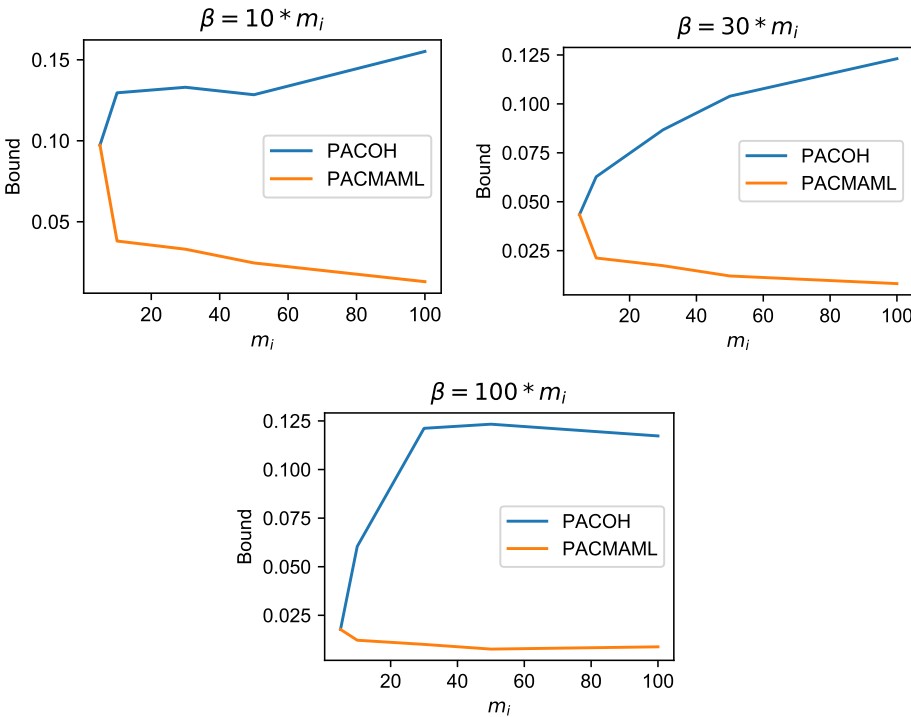

Figure 7: Comparison of the values of PACOH and PACMAML bound for $\beta/m_i \in \{10, 30, 100\}$. $\alpha$ for each $m_i$ for PACMAML is set to the optimal value according to Fig. 4.

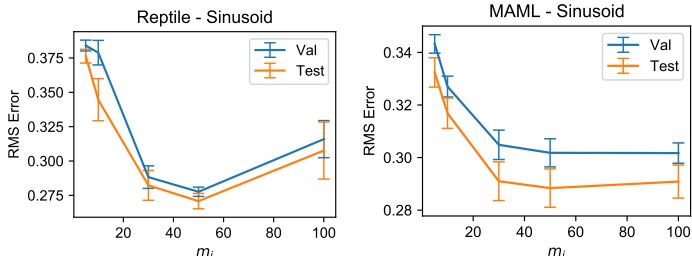

Figure 8: Mean and standard error of the validation and the test result for Reptile and MAML on Sinusoid. The results are obtained from cross-validation. The error bars in the figures represent the standard errors.

# E   Experiment Details of Image Classification

For most hyperparameters, we followed the same default values as in [9]. In Table 5, we listed the hyperparameters that we did grid search, and their chosen value based on the meta-validation performance. For the inner learning rate, the search space was $\{0.1, 0.03, 0.001, 0.003\}$ for FOMAML, MAML, and PACMAML; the search space was $\{0.1, 0.03, 0.001, 0.003, 0.001, 0.0003, 0.0001\}$ for BMAML and PACOH. For the meta-learning rate, we used the default 0.001 for FOMAML, MAML and PACMAML; and searched over $\{0.001, 0.0003, 0.0001, 0.00003\}$ for BMAML and PACOH. For $\alpha$, we searched over $\{10, 1.0, 0.1\}$ for BMAML, PACOH, PACMAML. We also tried two gradient descent methods in the inner loop: Vanilla GD and ADAGRAD . We found that FOMAML and MAML worked better with Vanilla GD; while BMAML, PACOH and PACMAML worked better with ADAGRAD. $\sigma^2$ was fixed to 1 for PACOH and PACMAML. The number of task per batch was 4 and the network filter size was 64. The total number of meta-training iterations was 60000 for all algorithms. We ran these tasks with 1 NVIDIA P100 GPU per job and each job takes about 2-3 hours to finish.

| $m_i$ | Hyper-parameter | FOMAML | MAML | BMAML | PACOH | PACMAML |
|---|---|---|---|---|---|---|
| | outer learning rate | 0.001 | 0.001 | 0.0001 | 0.0001 | 0.001 |
| 10 | inner learning rate | 0.1 | 0.1 | 0.003 | 0.01 | 0.03 |
| | $\alpha$ | - | - | 1.0 | 10 | 1.0 |
| | outer learning rate | 0.001 | 0.001 | 0.0001 | 0.0001 | 0.001 |
| 20 | inner learning rate | 0.03 | 0.1 | 0.003 | 0.003 | 0.01 |
| | $\alpha$ | - | - | 0.1 | 1.0 | 1.0 |
| | outer learning rate | 0.001 | 0.001 | 0.0001 | 0.0001 | 0.001 |
| 40 | inner learning rate | 0.03 | 0.03 | 0.003 | 0.003 | 0.01 |
| | $\alpha$ | - | - | 0.1 | 1.0 | 10 |
| | outer learning rate | 0.001 | 0.001 | 0.0001 | 0.0001 | 0.001 |
| 80 | inner learning rate | 0.03 | 0.03 | 0.0003 | 0.003 | 0.01 |
| | $\alpha$ | - | - | 0.1 | 1.0 | 1.0 |

Table 5: The final hyper-parameters of the algorithms in the Mini-imagenet task.

# F  Experiment Details of Natural Language Inference

We fixed $\sigma^2 = 0.0004$, which equals to the variance of the BERT parameter initialization. The hyper-parameter $\alpha$ is decided by a grid search over $\left\{10^2, 10^3, 10^4, 10^5, 10^6, 10^7\right\}$ based on the performance on the meta-validation dataset. The inner loop learning rate is $0.001$ for all algorithms. We used 50-step Adagrad optimizer in the inner-loop because it has automatic adaptive learning rate for individual variables which is beneficial for training large models. For the outer-loop optimization, we used the ADAM optimizer with learning rate $10^{-5}$. The final hyperparameters are reported in Table 6. In the few-shot learning phase, we ran the ADAM optimizer for 200 steps with learning rate $10^{-5}$ on the adaptable layers. We ran the tasks with 16 TPUs(v2) per job.

| Hyper-parameter | MAML | BMAML | PACOH | PACMAML |
|---|---|---|---|---|
| inner learning rate | 0.001 | 0.001 | 0.001 | 0.001 |
| $v$ | 12 | 12 | 12 | 11 |
| $m_i'$ | 32 | 64 | 256 | 64 |
| $m_i$ | 256 | 256 | 256 | 256 |
| $\alpha$ | - | $10^3$ | $10^4$ | $10^4$ |
| tasks per batch | 1 | 1 | 1 | 1 |
| meta-training iteration | 10000 | 10000 | 10000 | 10000 |

Table 6: The final hyper-parameters in the NLI tasks.

In Table 7 we report the detailed classification accuracy on the 12 NLI tasks with their standard errors.

| Task name | $N$ | $k$ | MAML | BMAML | PACOH | PACMAML |
|-----------|-----|-----|------|-------|-------|---------|
| CoNLL | 4 | 4 | 63.0±1.4 | 61±2.3 | 62.1±2.2 | 68.8±1.6 |
|  |  | 8 | 74.1±1.8 | 68±1.9 | 74.9±1.2 | 79.5±1.1 |
|  |  | 16 | 81.6±0.6 | 77.9±1.4 | 83±0.7 | 84.5±0.6 |
| MITR | 8 | 4 | 51.3±1.8 | 47.5±1.9 | 55.9±1.6 | 60.6±1 |
|  |  | 8 | 69.1±2.1 | 64.2±1.3 | 71.8±0.8 | 70.9±1 |
|  |  | 16 | 78.7±1.1 | 72.2±1.3 | 78.1±0.6 | 80±0.6 |
| Airline | 3 | 4 | 60.1±2.0 | 53±2.7 | 60.1±3.1 | 60.5±1.9 |
|  |  | 8 | 64.7±2.7 | 67.4±2.2 | 65±1.5 | 65.4±1.7 |
|  |  | 16 | 68.4±2.2 | 66.7±2.6 | 69.6±1.3 | 69.9±1.1 |
| Disaster | 2 | 4 | 56.3±0.5 | 58.7±3.1 | 58.7±2.6 | 63.3±1.3 |
|  |  | 8 | 61.5±0.7 | 64.1±2.3 | 64.1±2.4 | 63.9±2.9 |
|  |  | 16 | 67.7±0.4 | 69.4±2.0 | 71.3±1.7 | 71.1±1.6 |
| Emotion | 13 | 4 | 13.7±2.1 | 13.9±0.5 | 13.8±0.5 | 13.7±0.7 |
|  |  | 8 | 15.8±1.9 | 14.6±1.1 | 15±0.6 | 15.8±0.6 |
|  |  | 16 | 16.7±0.9 | 15.6±0.7 | 17.2±0.7 | 16.8±0.5 |
| Political Bias | 2 | 4 | 58±2.1 | 58±2.0 | 58.8±2.6 | 59.9±2.1 |
|  |  | 8 | 60.7±1.9 | 61±1.9 | 62.1±1.5 | 62±1.9 |
|  |  | 16 | 64.6±0.9 | 63.5±1.2 | 63.8±1.2 | 66±1 |
| Political Audience | 2 | 4 | 52.2±0.9 | 54.9±0.7 | 53.1±0.9 | 53.4±1.3 |
|  |  | 8 | 56.1±1.5 | 55.9±1.1 | 56±1.3 | 56±1.2 |
|  |  | 16 | 56.5±1.2 | 56.9±1.3 | 60±0.9 | 59.6±1 |
| Political Message | 9 | 4 | 18.9±0.8 | 17.4±0.6 | 19.2±0.7 | 19.3±0.6 |
|  |  | 8 | 22.3±0.7 | 19.3±0.8 | 22.3±0.6 | 22.6±0.5 |
|  |  | 16 | 24.3±0.8 | 21.6±0.4 | 24.9±0.4 | 25.5±0.8 |
| Rating Books | 3 | 4 | 58.7±2.1 | 56.2±2.8 | 59±2.3 | 56.8±3 |
|  |  | 8 | 61.3±2.7 | 55.1±2.7 | 64.2±2 | 61.6±1.5 |
|  |  | 16 | 62±1.3 | 66.6±2.1 | 63±2.1 | 60.4±2.7 |
| Rating DVD | 3 | 4 | 49.5±3.0 | 53.7±2.7 | 53.7±2.1 | 52.4±1.5 |
|  |  | 8 | 53.2±1.6 | 51.8±2.4 | 54.7±2 | 56±2 |
|  |  | 16 | 54.7±1.2 | 57.2±1.5 | 55.4±1.3 | 60±1.4 |
| Rating Electronics | 3 | 4 | 46.9±3.1 | 44.6±1.9 | 53.3±1.7 | 52.4±2 |
|  |  | 8 | 52.5±1.6 | 54.1±1.6 | 55.6±2 | 56.1±1.3 |
|  |  | 16 | 54.7±1.8 | 56.6±1.8 | 57.5±1.5 | 58.2±0.7 |
| Rating kitchen | 3 | 4 | 49.9±2.4 | 48.3±2.1 | 57.9±1.3 | 57.8±2 |
|  |  | 8 | 50.9±2.8 | 49.5±3.1 | 52.3±2.2 | 58.3±1.5 |
|  |  | 16 | 58.7±1.5 | 54.2±1.8 | 54.8±1.8 | 58.1±2.5 |
| Overall average | - | 4 | 48.21 | 47.27 | 50.47 | 51.58 |
|  |  | 8 | 53.52 | 52.08 | 54.83 | 55.68 |
|  |  | 16 | 57.38 | 56.53 | 58.22 | 59.18 |

Table 7: Classification accuracy and standard error on the 12 NLI tasks.