# OpenReview forum: "Bridging the Gap Between Practice and PAC-Bayes Theory in Few-Shot Meta-Learning"
_NeurIPS.cc/2021/Conference — NeurIPS 2021 Poster_

### Official Review · Reviewer_pWqh · 2021-07-11

**Rating:** 7
**Confidence:** 4

**Summary:**

This paper studies the Few-shot meta-learning setting according to PAC-Bayes theory. The authors focus on a context where the number of examples from source tasks and the target ones do not follow the same distribution. The authors develop two PAC-Bayes bounds tailored for the few-shot learning setting. Additionally, two existing and famous few-shot learning algorithms (MAML and Reptile) can be derived from the proposed bounds.
In that sense, the work aims at reducing the gap between theory and practice by proposing a theory that can justify some existing and efficient algorithms. A new algorithm is also derived from the proposed framework PACMAML. An experimental evaluation on a synthetic problem and on existing benchmarks is presented.

**Limitations And Societal Impact:**

In the conclusion, the authors mention a limitation related to the fact that the method does not consider the domain shift setting.

This was not mentioned in the paper, but this work is mainly fundamental and is not subject directly to negative social impact.


**Main Review:**

Originality
--------
Understanding few-shot learning methods is still challenging and this work participates to a better understanding of this framework, which represents an important value.
Existing PAC-Bayesian settings for few-shot/meta-learning do not handle the difference in terms of training size between source and target tasks, so there is a novel aspect here that has the advantage to be more realistic for the few-shot learning setting.
Another property of the framework is to allow to interpret MAML and REPTILE, which also new and offers some justification for these two algorithms, even though this may benefit from more discussion.
An algorithm is derived, using a base strategy borrowed from the PACOH paper, anyway a specific novel formulation is derived.

Overall, the novelty of the paper is rather fair.


Quality
------
The results are solid and well justified. All claims are justified by proofs and derivations, sometimes included in the Appendix.
The experimental evaluation is rather large and solid.


Clarity
------
The paper is globally clear and convincing.
I had a problem for interpreting the term $\Delta_\lambda$ in Thm3, I would appreciate if the authors can comment more on it.
Maybe the assumptions on T and \hat{T} could be more precisely discussed to avoid any misunderstanding.

Significance
----------
I think that the paper provides a significant contribution for studying the few-shot learning in PAC-Bayes, mainly in three aspects: (i) a framework that takes into account the different sizes between observed and target tasks, (ii) the capacity of the framework to allow an interpretation of two existing algorithms, (iii) the derivation of a new algorithm that provides better results in comparison to other approaches ) related to Pac-Bayes though.

The significance outside the PAC-Bayesian community is not clear but at least the work contributes to the understanding of few-short problem which is important.

Some methodological aspects and results could be more discussed anyway.


Comments
-----------

-line 130: Definition and differences of $T$ and $\tilde{T}$ should require a bit more precision on the expected differences in terms of tasks environment. Is it just that the samples are draws with less examples, or can we imagine more (structural) differences?
According to assumptions of Thm3, line 132/133, if I understood correctly, the environment can output the same distributions, so in other words, the same tasks, but the difference is that in T the expected number of observed samples is smaller.

line 136: I am not sure to understand/interpret correctly the $\Delta_\lambda$ term. From the hypothesis of line 132/133, I understand that the 2 environments provide the same tasks, in expectation, so I would have appreciated more precision in order to explain  why $R(P,T)-R(P,\tilde{T})$ should not be 0 in expectation? One explanation coud that the tasks weights are different between the two environnements (related to my question above but this probably not the case), but my interpretation is that when using smaller samples the base algorithms would produce less powerful base classifiers which cannot be compensated by the posterior, but I may have  wrongly interpreted the notation. I would appreciate if the authors could comment on it.

line 137, first equation: you expect that the number of instances samples in the observed tasks tends to the observed harmonic mean?

-Table 1: the results seem to correspond to accuracy and not generalization error (since the true error cannot be estimated anyway), same for Table2.

-Fig 2, line 279. The U-shaped can indeed shows the beginning of an overfitting situation where the learners starts overfitting on the source tasks. That being said, this could be explained by the fact that the base learners are not chosen, parameterized or regularized correctly for the problem considered. This may indicates that some information are still missing in the proposed bound.

-The analysis of the generalization bounds is deferred to Appendix with the conclusion that PACMAML exhibits better generalization bound than the competitor PACOH in the context of the synthetic experiments. I would be interested in knowing the bounds for MAML or REPTILE for example, to see if the bounds can be informative to perform algorithm selection for some problems. Note also that showing bounds for the other tasks is interesting.

-The bounds provides PAC-Bayesian guarantees and allows to deduce some algorithms, which is good. Maybe something that is missing is to have some information about a structure on the posterior, for example maybe some diversity should be observed on the base learners and when this occurs maybe the guarantees can be better. Maybe extensions that would incorporate such a property would be interested.

-For PACMAML, a particular formulation of the Gibbs posterior is used. This is expected since otherwise one cannot expect to have a tractable approach. However, I would expect a little discussion on the expressiveness of this choice and maybe some possible other alternatives.

In practice a Monte-Carlo sampling is used, but this implies probably high computational costs, what happened on existing image and natural language benchmarks?

-The bound proposed by the paper uses the frameworks of Pentina et al. and Rothfuss et al. that rely on the McAllester version of a PAC-Bayes bound. However, there exist tighter version, like the Seeger one, which may help to have even tighter bound, but at a price of a higher complexity and to derive new specific studies.

-line 157 and also l166, the distribution on which expectation of m'_i is not indicated, should you define an other environment that depends on S_i and D_i.

-A point that is a bit puzzling is that since the framework can be interpreted for deriving the algorithms MAML and REPTILE, one can then learn with these algorithms while optimizing a PAC-Bayes bound. I would then expect a better behavior for these 2 approaches.
That being said, if we reinterpret these two algorithms with the proposed framework, can we explain by the bound why these algorithms would perform worse than PACMAML? Is it because of the optimization problem that is significantly harder to optimize, or is it because the considered Gibbs posterior is not well adapted for these algorithms, or maybe there is another reason?


-----
After rebuttal
-----
Thank you for the answers provided.

On the remark about information on the posterior, maybe considering bounds that involve information on the diversity of the voters can be interesting, such as:
*the C-bound: Pascal Germain, Alexandre Lacasse, François Laviolette, Mario Marchand, and Jean-Francis Roy.
Risk bounds for the majority vote: From a PAC-Bayesian analysis to a learning algorithm. Journal
of Machine Learning Research, 16, 2015.
*or 2nd order bounds: Andres R. Masegosa, Stephan S. Lorenzen, Christian Igel, Yevgeny
Seldin. Second Order PAC-Bayesian Bounds for the Weighted Majority Vote.
NeurIPS 2020.


**Time Spent Reviewing:**

5

---

> ### Author Response · Authors · 2021-08-07
> **Author Response to Reviewer pWqh**
>
> Thank you for your questions. Please see our response below.
>
> Q: line 130: Definition and differences of $T$ and $\tilde{T}$ should require a bit more precision on the expected differences in terms of task environment. Is it just that the samples are draws with less examples, or can we imagine more (structural) differences? According to assumptions of Thm3, line 132/133, if I understood correctly, the environment can output the same distributions, so in other words, the same tasks, but the difference is that in T the expected number of observed samples is smaller.
>
> A: Your understanding is correct. In our definition, the task environment $T$ contains two components $(D, m)$. In this paper, we do not consider the case where the data distributions $D$ of $T$ and $\tilde{T}$ are different, but only focus on the case where their numbers of samples $m$ are different. Nevertheless, the bound in Thm3 can be used for $\tilde{T}$ where its data distribution $D$ is different from $T$ as well.
>
> Q: line 136: I am not sure to understand/interpret correctly the $\Delta_\lambda$ term. From the hypothesis of line 132/133, I understand that the 2 environments provide the same tasks, in expectation, so I would have appreciated more precision in order to explain why $R(P,T)−R(P,\tilde{T})$ should not be 0 in expectation? One explanation could that the tasks weights are different between the two environnements (related to my question above but this probably not the case), but my interpretation is that when using smaller samples the base algorithms would produce less powerful base classifiers which cannot be compensated by the posterior, but I may have wrongly interpreted the notation. I would appreciate it if the authors could comment on it.
>
> A: Your second interpretation is correct. Although the two environments have the same data distribution, the task drawn from $T$ contains fewer examples than the ones from $\tilde{T}$. In Thm-3, this means that the base-learner during meta-training sees more data examples than the base-learner during meta-testing. This is what introduces the $\Delta$ term.
> Thm-4 removes this gap between meta-training and meta-testing, so that the $\Delta$ term disappears. We will emphasize the difference between $T$ and $\tilde{T}$ and clarify this better in the final version of the paper.
>
> Q: line 137, you expect that the number of instances samples in the observed tasks tends to the observed harmonic mean?
>
> A: The main point here is that the meta-training tasks have far more samples $m_i$ than the meta-testing task samples $m$. In this sentence, we try to say that the expected value of $m_i$ and their harmonic mean $\tilde{m}$ are the same or similar. We will remove the equality and only retain the relation of $E[\tilde{m}] \gg E[m]$ in the final version.
>
> Q: Table 1: the results seem to correspond to accuracy and not generalization error (since the true error cannot be estimated anyway), same for Table2.
>
> A: You are right. We will correct this mistake. Thank you!
>
> Q: Fig 2, line 279. The U-shaped can indeed show the beginning of an overfitting situation where the learner starts overfitting on the source tasks. That being, this could be explained by the fact that the base learners are not chosen, parameterized or regularized correctly for the problem considered. This may indicate that some information is still missing in the proposed bound.
>
> A: We reported each point in Figure 2 by cross-validation and a grid search of hyperparameters so as to minimize the possibility of the inferior performance due to poor parameter choice. Please refer to D.2, D.3, D.5 for the details.
>
> Q: The analysis of the generalization bounds is deferred to Appendix with the conclusion that PACMAML exhibits better generalization bounds than the competitor PACOH in the context of the synthetic experiments. I would be interested in knowing the bounds for MAML or REPTILE for example, to see if the bounds can be informative to perform algorithm selection for some problems. Note also that showing bounds for the other tasks is interesting.
>
> A: Thank you for the suggestions. The PAC-Bayes bounds corresponding to MAML and REPTILE are shown in Eq.(7) subject to a constant. We will investigate this to see if this brings any useful information. Please also see our response to Reviewer Jt6H for a related question about the bounds of other tasks.
>
> Q: The bound provides PAC-Bayesian guarantees and allows to deduce some algorithms, which is good. Maybe something that is missing is to have some information about a structure on the posterior, for example maybe some diversity should be observed on the base learners and when this occurs maybe the guarantees can be better. Maybe extensions that would incorporate such a property would be interesting.
>
> A: Thanks for this interesting suggestion. If you could provide related references, we will look into this in the future.
>
> Q: For PACMAML, a particular formulation of the Gibbs posterior is used. This is expected since otherwise one cannot expect to have a tractable approach. However, I would expect a little discussion on the expressiveness of this choice and maybe some possible other alternatives. In practice, Monte-Carlo sampling is used, but this implies probably high computational costs. What happened to existing image and natural language benchmarks?
>
> A: Besides the exponential family, another feasible choice would be the Dirac measure of MAML. Other choices that we are aware of may be Bayesian mixture models, but they are significantly more expensive.
> Regarding the image and natural language benchmarks, in order to fairly compare with MAML and other methods for large-scale meta-learning, we used only one MCMC sample to approximate $Q(w, S)$ (mentioned in line 292), so that the computational cost is similar or better (than the full MAML due to its high-order derivative). Our experiments indicate that using 5 samples may further improve some results by about 1% but at a higher computational cost.
>
> Q: The bound proposed by the paper uses the frameworks of Pentina et al. and Rothfuss et al. that rely on the McAllester version of a PAC-Bayes bound. However, there exist tighter versions, like the Seeger one, which may help to have even tighter bound, but at a price of a higher complexity and to derive new specific studies.
>
> A: Thank you for the suggestions. We will look into this in the future.
>
> Q: line 157 and also l166, the distribution on which expectation of $m'_i$ is not indicated, should you define another environment that depends on $S_i$ and $D_i$.
>
> A: Here we mean that $m’_i$ follows the same distribution and expectation of $m$.
> We will clarify this in the final version.
>
> Q: A point that is a bit puzzling is that since the framework can be interpreted for deriving the algorithms MAML and REPTILE, one can then learn with these algorithms while optimizing a PAC-Bayes bound. I would then expect a better behavior for these 2 approaches. That being said, if we reinterpret these two algorithms with the proposed framework, can explain by the bound why these algorithms would perform worse than PACMAML. Is it because of the optimization problem that is significantly harder to optimize, or is it because the considered Gibbs posterior is not well adapted for these algorithms, or maybe there is another reason?
>
> A: Both MAML and REPTILE restrict the base-learner to be the Dirac measure. PACOH and PACMAML use the Gibbs distribution for the base-learner, which is Bayes-optimal for PACOH. We therefore expect PACOH to behave better than REPTILE. In terms of MAML, the full MAML algorithm relies on high-order derivatives, which appear to be hard to optimize for very deep models (e.g. discussed in [4]), and the FOMAML algorithm completely throws away those high-order information. In comparison, the PACMAML algorithm has high-order information in the first-order format (2nd term of Eq(14)), which explains why it outperforms both FOMAML and MAML.

---

### Official Review · Reviewer_587T · 2021-07-12

**Rating:** 8
**Confidence:** 3

**Summary:**

Previous PAC-Bayesian Meta-learning bounds assume the number of samples per task follows the same distribution in the meta-training and in meta-testing.  This paper relaxes this assumption, allowing for a shift in the number-of-samples distribution from meta-training to meta-testing (Thm 3).
In Thm 4, the paper suggests a learning strategy that omits the extra distribution shift of Thm 3, by using a smaller data set.
The paper shows how the popular meta-learning algorithms, Reptile, and MAML can be justified using the PAC-Bayes framework, and also introduces a new algorithm called PACMAML.  Experimental results show the method outperforms other algorithms on several few-shot benchmarks.

**Limitations And Societal Impact:**

The limitations are discussed in the main review. There is no potential negative societal impact.

**Main Review:**


Strengths
- The paper is well written. The structure is logical and easy to follow.
- Interesting observation from theory - tighter meta-test bound can sometimes be achieved if the base learner uses FEWER samples  (the same number of samples as in meta-training), this seems counter-intuitive, but helps to mitigate the distributional shift in the distribution over the number of samples.
- Experimental results show the suggested algorithm has favorable performance in few-shot tasks.
**************************************************************
Weakness
- There is an issue that is not addressed. The empirical error increases if fewer samples are used (since we have a base-learner that was trained with fewer samples).  Is it ensured that this increase does not negate the reduction in the complexity term? The paper does not seem to discuss this issue.
-The derivation of the gradient estimators is not clear. For example: (1) below line 602 in the appendix, there is an inequality, but the final result is equality. (2) I couldn’t understand the transitions below line 603.
- The sampling from the distribution $Q(w, S)$ is not trivial (since it involves an exponent of the empirical loss). However, the paper doesn’t elaborate on the sampling method used besides referencing SGLD and SVGD. Since there are only approximated methods for sampling, with advantages and disadvantages, I believe this part should be investigated more in-depth. Furthermore, there is no pseudocode of the full algorithm, and so readers can not recarate the results.
**************************************************************
Questions
- Line 317 - “the base-learner is only trained on S’ and the meta-learner can learn from the unseen examples in S and therefore no overfitting happens”. This sentence is not clear. How is it guaranteed that there is no overfitting?
**************************************************************
Minor comments:
- Line 88 - possible typo -  $S_i \sim D$ instead of  $S_i \in D$
- Line 130 - \tilde{T} is undefined
- Lines 137-138 are not clear.
- Line 225, I think REINFORCE is not the name of the method used since it usually uses the log-derivative trick.
- Figure 2 lacks error bars.


**Time Spent Reviewing:**

9

---

> ### Author Response · Authors · 2021-08-07
> **Author Response to Reviewer 587T**
>
> Thank you for your questions. Please see our response below.
>
> Q: There is an issue that is not addressed. The empirical error increases if fewer samples are used (since we have a base-learner that was trained with fewer samples). Is it ensured that this increase does not negate the reduction in the complexity term? The paper does not seem to discuss this issue.
>
> A: The empirical error (1st term of eq(6) or eq(5)) depends on two factors: the base-learner $Q(S_i’, P)$ and the meta-learner $\mathcal{Q}$. For the same meta-learner $\mathcal{Q}$, it is true that the base-learner of Thm3 should get lower empirical error. However, Thm4 may result in a better meta-learner $\mathcal{Q}$ than Thm3 (see our answer regarding ‘overfitting’).
> To demonstrate this empirically, In Fig.5 and Fig.6 of D.4, we decomposed the generalization bounds and plotted $W_1$ (for Thm3) and $W_2$ (for Thm4) which are equal to the 1st term (empirical error) + the 3rd term of eq(5) or eq(6). We can see that $W_1$ is actually larger than $W_2$. On the other hand, the dominant factor is still the $\Delta$ term of Thm3.
>
> Q: The derivation of the gradient estimators is not clear. For example: (1) below line 602 in the appendix, there is an inequality, but the final result is equality. (2) I couldn’t understand the transitions below line 603.
>
> A: (1) Line 603 is the gradient of the 2nd line of 602 (softmax value function), which is not the exact gradient of the 1st line of 602 but serves as a low-variance approximation. Therefore, the final result in 604 is an approximation of the true gradient of $W_2$, so we used $\simeq$ in line 604 as well as in Eq(14) in line 237.
> (2) We indeed skipped a few intermediate steps that lead to these results, and we will add them back for clarity. Briefly speaking, the 2nd/3rd lines of 603 are derived by exploiting the relation between $Q_i^{\alpha}$, $Q_i^{\beta}$ and the derivatives of exponential families. As for the 1st line of 604, the first term comes from the sum between the gradient of the first term of $W_2$ and the last line of 603; the second term comes from the first term of eq(25).
>
> Q: The sampling from the distribution $Q(w, S)$ is not trivial (since it involves an exponent of the empirical loss). However, the paper doesn’t elaborate on the sampling method used besides referencing SGLD and SVGD. Since there are only approximated methods for sampling, with advantages and disadvantages, I believe this part should be investigated more in-depth. Furthermore, there is no pseudocode of the full algorithm, and so readers can not recarate the results.
>
> A: The exponent of the empirical loss actually makes the inference easier, because $Q(w, S)$ become exponential families and SGLD and SVGD are based on the gradient of the logarithm of the unnormalized posterior (see [14, 27] or https://en.wikipedia.org/wiki/Stochastic_gradient_Langevin_dynamics). In order to fairly compare with MAML, we used only one sample to approximate $Q(w, S)$. We tried both SGLD and SVGD (which reduces to SGD in the log space for the one sample case) in our image and natural language experiments, and they yielded similar results. Using 5 samples may further improve some results by 1%. We will include the pseudocode in the appendix.
>
> Q: “the base-learner is only trained on $S’$ and the meta-learner can learn from the unseen examples in $S$ and therefore no overfitting happens”. This sentence is not clear. How is it guaranteed that there is no overfitting?
>
> A: In the PACOH framework, both the base-learner and the meta-learner are trained over the same dataset $S$. When the base-learner achieves zero loss over $S$, there is essentially nothing left for the meta-learner to learn. This is what we mean by “overfitting” (of the base-learner to the training data of the meta-learner). Importantly, if the meta-learner is unable to learn, then it would not learn representations that are useful for other tasks.
> More specifically, in eq(12) of PACOH, when the task $S$ is simple (e.g. 5 way classification for 10 data points), $w$ from the base-learner $Q(w, S)$ alone may already perfectly fit the data, yielding $L(p+w, S)$ close to 0. This would make the gradient in (12) close to 0 and leave the meta-parameter $p$ nothing to learn.
> In contrast, this type of overfitting cannot happen in the PACMAML framework, because the base-learner is trained over a subset $S'$ of the data, and the meta-learner can still learn from the unseen samples in $S / S'$. That is, in the first term of (14), the base-learner adapts on $Q(w, S’)$ but the loss $L(p+w, S)$ is computed over $S$ and would not be close to 0 because $S \neq S’$ contains previously unseen samples.

---

> > ### Comment · Reviewer_587T · 2021-08-17
> > **response to rebuttal**
> >
> > I thank the authors for their detailed answers.
> > After reading the discussion, I feel the authors had adequate answers to the reviewers' concerns.
> > I think the paper raises original points and studies them thoroughly.
> > I raised the score accordingly.

---

### Official Review · Reviewer_Jt6H · 2021-07-20

**Rating:** 8
**Confidence:** 2

**Summary:**

Existing meta-learning techniques assume that the number of samples for teach task are drawn from the same distribution.  Theorem 3 shows that a naive decoupling of the number of samples introduces a term to the gap that penalizes additional samples.  The paper bypasses this issue by introducing a fix-sized subset that is used to train a base model.  The additional samples are then used to evaluate the empirical risk, thus removing the aforementioned unsatisfactory term.  The MAML and Reptile algorithms are recovered by considering the derived bound and normal priors along with both delta posteriors (ie MAP solutions) and normal posteriors.  Results are reported on synthetic regression, image classification, and product review classification.

**Limitations And Societal Impact:**

There are no negative societal implications.  The major limitations are identified in the text: "One major limitation of the work is that we do not take into account a data domain shift (e.g. [11]), which is often present in practice."

**Main Review:**

### Positives

While I am not an expert in meta-learning or PAC-Bayes, this work seems well motivated (fixing the issue of uniformity in the sample sizes of previous PAC bounds), well executed / technically correct, and novel.


### Negatives

I was confused a bit by the experimental section.  I thought that the paper would experimentally validate the derived bound, comparing it not only to the empirical generalization error but also the bounds previously derived for meta-learning.  This is done for the synthetic data in Figure 1 but no where else.  Why aren't the bounds empirically studied?

**Time Spent Reviewing:**

2

---

> ### Author Response · Authors · 2021-08-07
> **Author Response to Reviewer Jt6H**
>
> Thank you for your questions. Please see our response below.
>
> Q: I thought that the paper would experimentally validate the derived bound, comparing it not only to the empirical generalization error but also the bounds previously derived for meta-learning. This is done for the synthetic data in Figure 1 but nowhere else. Why aren't the bounds empirically studied?
>
> A: Empirical estimation of the bound is generally very computationally expensive for large models without a closed form (e.g. the expected $\log Z$ term for classification tasks, the $\Delta_\lambda$ term) and subject to high variance with MCMC approximations. Therefore, almost no existing PAC-Bayes papers on meta-learning (e.g. [3, 18, 22]) that we know of empirically evaluate their bounds. However, compared to the previous papers, we do empirically study the bounds over the synthetic Sinusoid regression task by using Gaussian processes which do provide closed-form formulas (see D.1) and empirically validate the advantage of our new bound in Thm 4.

---

### Decision · Program_Chairs · 2021-09-27

**Decision:**

Accept (Poster)

**Comment:**

All reviewers and myself agree on the interest of the topic for the NeurIPS community, on the significance of the results, and on the clarity of the contributions. The discussion phase allowed to satisfactorily address the few concerns expressed by the reviews. As pointed out by a reviewer, the submission provides valuable insights on meta-learning generalisation bounds when the base learner has access to small amounts of data, which in some cases leads to tighter bounds. This observation helps to mitigate the distributional shift in the distribution over the number of samples.